# Late Eocene to early Oligocene productivity events in the proto-Southern Ocean and correlation to climate change

Gabrielle Rodrigues de Faria[1, 2], David Lazarus[1], Johan Renaudie[1], Jessica Stammeier[3], Volkan Özen[1, 2], Ulrich Struck[1, 2]

[1]Museum für Naturkunde, Leibniz Institute for Evolution and Biodiversity Science, Invalidenstraße, 43, Berlin, 10115, Germany

[2]Freie Universität Berlin, Institute for Geological Sciences, Malteserstraße 74-100, Berlin, 12249, Germany

[3]GFZ German Research Centre for Geosciences, Telegrafenberg, Potsdam, 14473, Germany

*Correspondence to*: Gabrielle Rodrigues de Faria (gabrielle.faria@mfn.berlin)

**Abstract.** The Eocene-Oligocene transition (EOT, ca 40-33 Ma) marks a transformation from a largely ice-free to an ice-house climate mode that is well recorded by oxygen stable isotopes and sea surface temperature proxies. Opening of the Southern Ocean gateways and decline in atmospheric carbon dioxide levels have been considered as factors in this global environmental transformation and the growth of ice sheets in Antarctica during the Cenozoic. A more comprehensive understanding is still needed of the interplay between forcing versus response, the correlation among environmental changes and the involved feedback mechanisms. In this study, we investigate the spatio-temporal variation in export productivity using biogenic Ba (bio-Ba) from Ocean Drilling Program (ODP) Sites in the Southern Ocean, focusing on possible mechanisms that controlled them as well as the correlation of export productivity changes to changes in the global carbon cycle. We document two high export productivity events in the Southern Ocean during the late-Eocene (ca. 37 and 33.5 Ma) that correlate to proposed gateway-driven changes in regional circulation, and to changes in global atmospheric $p$CO$_2$ levels. Our findings suggest that paleoceanographic changes following Southern Ocean gateway openings, along with more variable increases in circulation driven by episodic Antarctica ice sheet expansion, enhanced export production in the Southern Ocean from the late Eocene through basal Oligocene. These factors may have played a role in episodic atmospheric carbon dioxide reduction, contributing to Antarctic glaciation during the Eocene-Oligocene transition.

## 1 Introduction

### 1.1 Late Eocene Events as Precursor to Antarctic Eocene/Oligocene Boundary Glaciation

The Eocene-Oligocene transition (EOT, ~ 40-33 Ma) is the most important climatic interval of the Cenozoic era (Westerhold et al., 2020). This interval involves profound transformations in environmental conditions including the onset of continental-scale Antarctica glaciation at the Eocene-Oligocene boundary (Shackleton & Kennett, 1975, Zachos et al., 1996, Coxall et al., 2005), sea-level fall (Houben et al., 2012) and global cooling (Prothero and Berggren, 1992; Liu et al., 2009; Bohaty et al., 2012; Hutchinson et al., 2021) as evidenced by a global shift in oxygen isotope records from biogenic calcium carbonate (>1‰; Zachos et al. 2001; Coxall et al. 2005; Bohaty et al., 2012; Westerhold et al., 2020). A positive deep-sea carbon isotope excursion of up to 1‰ (Zachos et al., 2001, Coxall et al,. 2005; Coxall and Wilson, 2011; Westerhold et al., 2020) and a change from a shallow (~3.5 km) to a deeper (~4.5 km) calcite compensation depth (CCD) (Coxall et al., 2005; Rea and Lyle, 2005; Pälike et al., 2012; Dutkiewicz and Müller, 2021;

Taylor et al., 2023) have also been observed and indicate that the carbon cycle played an important role in the changes observed during the transition, although the mechanisms that caused the carbon cycle perturbation are still unsolved.

Carbon cycling acts through a variety of feedback mechanisms. Even though it is well recognized that changes in atmospheric carbon dioxide ($CO_2$) impact the Earth's climate because of its large effect on temperature (Arrhenius, 1896; IPCC, 2021), the mechanisms controlling $CO_2$ levels over long timescales are still a matter of debate. A simplified carbon cycle theory suggests that the level of $CO_2$ is expected to remain in a steady state over multi millennial (>10 kyr) timescales (Berner et al., 1983; DeVries, 2022). This stability is attributed to a dynamic interplay of transfer of carbon between volcanic inputs, oceans, atmosphere and sequestration in marine sediments involving several mechanisms such as weathering, volcanism, and the ocean's biological pump.

However, the Earth's $pCO_2$ has, in fact, undergone substantial changes, from Glacial-Interglacial timescales (Sigman & Boyle, 2000) to long term changes during the Cenozoic (Beerling and Royer, 2011; Anagnostou et al., 2016), thus suggesting that the Earth's carbon system operates in a more complex way than proposed in the canonical model.

Changes in the Southern Ocean productivity are thought to have altered $pCO_2$ levels in the Earth's past, specifically on Glacial-Interglacial timescales (Archer et al., 2000; Sigman et al., 2010), although magnitude and timing remain debated. Changes in the late-Eocene Southern Ocean export productivity over 1 million years, as documented in our research, could potentially have impacted $CO_2$ levels, as hypothesized by Egan et al., (2013), based on Si isotope proxy data for Paleogene Southern Ocean diatom productivity.

The carbon cycle perturbation at the EOT provides an opportunity to understand climate-carbon cycle feedback (Zachos and Kump, 2005). The mechanisms proposed to explain such perturbations are processes operating gradually over long timescales, and may have had their origins in the middle to late Eocene. Preceding the abrupt change at the E/O boundary, the late Eocene was a period of gradual cooling and progressive $CO_2$ levels decrease (Lauretano et al., 2021). The events associated with this time period may have had substantial importance in pre-conditioning the climate system prior to the major climatic shift at the E/O boundary (Egan et al., 2013). The potential main drivers for the initiation of this global cooling and ice build-up in Antarctica are actively debated. Declining global atmospheric carbon dioxide concentrations, and the opening of Southern Hemisphere oceanic gateways, namely the Drake Passage (DP) and the Tasmanian Gateway (TG), are often proposed hypotheses to explain this transition (Coxall and Pearson, 2007; DeConto et al. 2008). The decline of carbon dioxide levels is an important factor in driving cooler temperatures, and has been suggested as the crucial factor in the EOT cooling and subsequent build-up of continental glaciers on Antarctica (DeConto and Pollard, 2003; Huber and Nof, 2006; Pagani et al., 2011). Atmospheric $CO_2$ partial pressure ($pCO_2$) decline has robust observational support. Atmospheric $pCO_2$ has been shown to decline through the Eocene, from ca 1400 p.p.m. at the Early Eocene Climate Optimum (EECO, ca 51 to 53 million years ago) to about 770 p.p.m. in the late-Eocene,.reaching a minimum of 550 ± 190 p.p.m. in the early Oligocene (Anagnostou et al. 2016). However, data is noisy, with significant variations among different proxies and thus details of the magnitude and timing are still unclear. Furthermore, the $CO_2$ threshold (~ 780 p.p.m.v) needed for the onset of Antarctica glaciation, is highly dependent on model boundary conditions while ice sheet model simulations reveal inter-model disagreement (Gasson et al., 2014). Therefore, it is crucial to approach this leading hypothesis with caution due to these uncertainties.

The tectonic opening of the Southern Ocean gateways is considered a mechanism contributing to the climatic shift because it allows the initiation of a circum-Antarctic flow, leading to the formation of the Antarctic Circumpolar Current (ACC) (Kennett, 1977; Barker, 2001; Scher and Martin, 2006; Toumoulin et al. 2020). This intense eastward flowing current is proposed in this hypothesis to impact the regional and global climate by preventing tropical heat of

low latitudes from reaching Antarctica, promoting the thermal isolation of Antarctica. Numerous ocean circulation model studies of this hypothesis have yielded conflicting results (Mikolajewicz et al., 1993; Najjar et al., 2002; De Conto & Pollard, 2003; Sijp et al., 2009; Goldner et al., 2014; Ladant et al., 2014; Inglis et al., 2015) but most of these earlier works were limited by unrealistic boundary conditions or other issues (Toumoulin et al., 2020, Hutchinson et al., 2021). Recent modelling circulation studies (e.g. Toumoulin et al., 2020; Sauermilch, et al. 2021) demonstrate the importance of the Southern gateway openings and the proto-ACC on ocean cooling in the Southern Hemisphere. Additionally, glaciation has itself a strong influence both on the circulation of the Southern Ocean and on global climate, via increased albedo, colder temperatures and increased latitudinal temperature gradients, and stronger zonal winds (Goldner et al., 2014). There is increasing evidence for at least partial, if transient Antarctic continental glaciation within the late Eocene (Scher and Martin, 2014), and thus this also needs to be considered in understanding how climate and ocean change developed within this period.

In addition to the above physical impacts, the ACC is associated with the development of the Southern Ocean fronts that contribute to upwelling-induced biological productivity (Chapman et al., 2020). Considering that the changes in the Southern Ocean circulation have the potential to affect export productivity and via its link to carbon sequestration in sediments, in removing $CO_2$ from the ocean-atmosphere system, the 'CO$_2$' hypothesis and the 'tectonic' hypothesis may be linked, via the influence that gateways may have had on Southern Ocean circulation, increasing export productivity enough to affect global $p$CO$_2$ (Egan et al., 2013). Therefore, evaluating export productivity patterns in the Southern Ocean across the Eocene-Oligocene and its relationship with circulation and decline of atmospheric carbon dioxide during this time period provide important information about possible climate feedbacks in this prominent climatic transition.

Many studies have shown variations in biological productivity during this time interval (Diester-Haass, 1995; Diester-Haass and Zahn, 1996; 2001, Salamy and Zachos, 1999; Diester-Haass and Zachos, 2003; Schumacher and Lazarus, 2004; Anderson and Delaney, 2005; Villa et al., 2014), pointing towards a productivity increase associated with ocean circulation changes that increased surface water nutrient availability (Diester-Haass 1992; Zachos et al 1996). However, existing studies have mostly focused on single sites, whose paleoceanographic history may reflect local rather than regional developments. A much broader spatial investigation is particularly important for understanding the influence of large-scale ocean circulation on this process. Moreover, the timing of productivity changes differs among the studies and different proxies, limiting our understanding of cause-and-effect relationship. This highlights the importance of well constrained age models and use of consistent paleoproductivity proxies.

Here, we reconstruct changes in export productivity in the Southern Ocean across the late Eocene and early Oligocene, and evaluate how the changes observed may be linked to ocean circulation changes and how they may have contributed to the climate changes observed at this interval. We utilize biogenic barium (bio-Ba) accumulation rates to measure marine export productivity. Bio-Ba is defined as the fraction of total barium that is not associated with terrigenous sources, sometimes referred to as *excess*-Ba (Dymond et al., 1992), and has been applied in several studies in the Paleogene (eg. Nielsen et al., 2003; Anderson and Delaney, 2005; Faul and Delaney, 2010). It is considered a relatively reliable proxy to estimate changes in paleoproductivity in the Southern Ocean. Newly generated carbon and oxygen stable isotope records from the same samples of our bio-Ba data further constrain possible causative mechanisms for the climatic shift at the EOT. We compare our export productivity proxy results to indicators of ocean circulation change, such as Neodymium (Nd) isotopes. Neodymium isotopes have emerged as a valuable geochemical water mass tracer (Piepgras and Wasserburg, 1982; Martin and Haley, 2000, Roberts et al., 2010), contributing significantly to our

understanding of the role of ocean circulation in the geological past. This proxy provides an opportunity to asses the
origin of water masses and reconstruct deep ocean circulation. Here, we use Nd isotope data to help understand gateway
opening, paleoceanographic changes and ice sheet history during the Eocene-Oligocene transition, and how these
changes may have influenced marine biological productivity. We also compare our productivity records to well
established proxies for the global carbon cycle, specifically $p$CO$_2$ and $\partial^{13}$C of benthic deep sea foraminifera.
Our understanding of the Eocene-Oligocene transition has been enhanced through modelling studies as they provide
means to compare several possible scenarios and generate 'data' for components of the system for which no direct
proxy data is available. However, all models are simplifications of complex, not fully understood systems; and are
dependent on parametrizations and calibrations to often sparse, noisy proxies that ground-truth model outputs. This
paper instead focuses on proxy evidence of the changes that occurred in this time interval.
**1.2. Paleoceanographic Setting**
The Southern Ocean (SO) today is an important part of the global ocean circulation and climate system, interconnecting
the Atlantic, Pacific and Indian Ocean basins, providing and thus inter-basin exchange of ocean properties and heat
(Rintoul et al., 2001). There are strong latitudinal gradients and seasonal changes in ocean properties which affect
surface water and export productivity, and thus this region's role in global carbon capture and sequestration. Low light
levels and, in higher latitudes, extensive sea ice limit productivity during the winter months. Deep surface mixed layers
over the large areas of the Southern Ocean, beyond the shallow stratification effects of meltwater near the sea ice edge,
also tend to limit productivity in spring through fall as plankton is mixed below critical thresholds of light availability
(Deppeler and Davidson, 2017). The relationship between mixed layer thickness and productivity however is complex
(Nelson and Smith, 1991; Li et al. 2021). Southern Ocean productivity is thus concentrated near the Antarctic
Circumpolar Current (ACC), the dominant current in the region. This current is the longest and strongest ocean current
on Earth. This complex circulation system is driven mainly by westerly winds, resulting in Ekman transport and
favouring deep water upwelling. This flow pattern is possible in the absence of land barriers and is governed by
bathymetry (Rintoul et al., 2001; Carter et al., 2008), while the strength of the current is driven by the strength and
location of the westerly winds, and thus, among other factors, the global latitudinal thermal gradient. The ACC is a key
component of the 'ocean conveyor belt', playing a role in the global transport of heat (Rintoul et al., 2001; Katz, et al.
2011). Moreover, this circumpolar current influences the strength of meridional overturning circulation and several
authors have proposed that this current is one of the main drivers of the Atlantic meridional overturning circulation
(AMOC) (Toggweiler and Samuels, 1995; Toggweiler and Bjornsson, 2000; Scher and Martin, 2006, Kuhlbrodt et al.,
2007, Scher et al., 2015, Sarkar et al., 2019).
The ACC is structured of multiple hydrological fronts, associated with specific water mass properties such as
temperature and salinity (Sokolov and Rintoul, 2009). Orsi et al., 1995 proposed the traditional view of Southern Ocean
fronts. It consists of the Subantarctic Front (SAF), the Antarctic Polar Front (APF) and the Southern ACC Front
(SACCF). Besides these main fronts, a Subtropical Frontal Zone (STFZ) can be found north of the ACC (Orsi et al.,
1995; Palter et al. 2013; Chapman et al., 2020). This frontal structure is fundamental to different processes that occur in
the region, such as the distribution of important nutrients through the exchange between deep and surface ocean, and the
exchange of tracers (Palter et al., 2013). Upwelling of Circumpolar Deep Water (CDW) brings nutrient-rich waters to
the surface towards the Polar Front Zone (PFZ) where Antarctic Surface Waters (AASW) sink to form Antarctic
Intermediate Water (AAIW), thereafter it extends into the Subantarctic Zone (SAZ) (Sarmiento et al., 2004) (Figure 1).
Wind-driven upwelling, that occurs within the Southern Ocean fronts, enhances biological productivity in these regions
(De Baar et al., 1995; Moore et al., 1999). More recently, upwelling related to ACC bathymetry has been found as an
important mechanism for establishing phytoplankton blooms in the SO (Sokolov and Rintoul, 2007). This complex
structure involving ACC fronts, westerlies and the bottom topography, makes the Southern Ocean a highly productive
region. Iron remobilisation has also been shown to occur due to latitudinal variations of the ACC (Kim et al., 2009),
inducing increases in productivity.
The conventional assumption is that the ACC structure began to develop during the Cenozoic, with the opening of the
Southern Ocean pathways between South America and Antarctica and the following formation of the Drake Passage
(DP) and, also between Australia and Antarctica that allows the Tasmanian Gateway (TG) opening. Removing these
geographic barriers permitted a gradual development of circumpolar flow (Toggweiler and Bjornsoon, 2000). The TG
opening to intermediate and deep waters occurred in the late Eocene, ca 35.5 Ma (Stickley et al., 2004). In contrast,
tectonic reconstructions for the timing of the Drake Passage opening remain controversial, ranging from the late Eocene
(ca 41 Ma; Scher and Martin, 2004, 2006) to the late Oligocene (ca 26 Ma, Barker and Thomas 2004, Hill et al., 2013)
for shallow water exchange. Some studies pointed to deep water exchange occurring as late as the earliest Miocene (ca
22-23 Ma, Barker 2001; Lyle et al., 2007). Even if the timing of the deepening of the Drake Passage is less well
constrained, a "proto-ACC" has been proposed as an earlier expression of the ACC and it is defined as a shallow-depth
circumpolar current (Scher et al., 2015, Sarkar, et al., 2019). Cramer et al. 2009 suggested that "proto-ACC" would
have played an important role in the ocean circulation changes that occurred in the Eocene. Furthermore, even a
relatively shallow proto-ACC would have strongly affected surface water phyto- and zooplankton (Lazarus and Caulet,
1994), and thus potentially the mechanisms of surface water productivity in the region.
Many climate model studies have contributed insights into the ocean structure and circulation of the late Eocene (e.g.
Huber et al. 2004, Huber & Not 2006, Sipj et al., 2011, Sijp et al., 2016, Elsworth et al., 2017, Baatsen et al., 2020;
Toumoulin, et al., 2020; Sauermilch et al., 2021; Nooteboom et al., 2022). Although some of these experiments have
shown that opening of gateways was not sufficient to have caused the global cooling recorded by proxies (DeConto &
Pollard, 2003, Huber et al. 2004, Huber & Not 2006, Sipj et al., 2011, Baatsen et al., 2020), they acknowledge that the
circulation patterns have changed during the Eocene. A recent model circulation experiment has demonstrated a
significant regional impact of the DP opening and its effects on ocean structure and dynamics even for shallow depths
(Toumoulin et al., 2020).
The organisation of Southern Ocean proto-oceanic fronts may have occurred during the late-Eocene as shown by
microfossil biogeographic data (Lazarus and Caulet, 1994; Cooke et al., 2002). This frontal system organization likely
played a role in major changes at that time period, including higher ocean productivity.
Evidence of significant events during the late Eocene highlights the importance of this period that preceded the
permanent glaciation in Antarctica. An interval of increasingly heavy global benthic oxygen isotope values in the late
Eocene, at ca 37 Ma have been interpreted to reflect pre-EOT glaciation and cooling, this episode is referenced as
PrOM event (Priabonian Oxygen isotope Maximum, Scher et al., 2014). Additional evidence for a prominent cooling
episode has been found during this time period (Anderson et al., 2011; Douglas et al., 2014). Despite uncertainties about
the nature and extent of the earliest ice in Antarctica, these changes imply that paleogeographic reconfiguration has
affected the late Eocene Antarctic climate and it is likely that a combination of processes favoured the development of
permanent glaciation in Antarctica.
Given the importance of changes during the Eocene-Oligocene time interval, especially the ACC development and its
frontal structure to the climate system and ecosystems, it is crucial to investigate the timing and magnitude of late
Eocene paleoceanographic changes in the Southern Ocean, and equally important to expand our understanding of the
implications of such changes on paleoproductivity and how these mechanisms are linked to a changing climate.
Our multiproxy approach and wide coverage allow us to test the hypotheses:
H1: Changes in ocean circulation patterns that took place during the late Eocene and early Oligocene (eg. development
of a proto-ACC and strengthening of AMOC) contributed to the increase in biological productivity in the Southern
Ocean.
H2: The export productivity increase that preceded the EOT may have been temporally correlated to, and thus may have
contributed to the drawdown of $p$CO$_2$.
First, we investigate the export productivity changes across the late Eocene to early Oligocene in three different regions
in the Southern Ocean. Then we compare our results to the paleo-circulation changes that occurred at the same time
period, and lastly compare our productivity records to the temporal pattern of change in Eocene-Oligocene $p$CO$_2$. We
conclude by summarising the implications of the changes in ocean circulation and the possible climate driving
mechanisms that led to the cooling of Earth.
**2 Materials and Methods**
**2.1 Site Descriptions**
We investigated sediment samples from 3 Ocean Drilling Program (ODP) Sites in the Southern Ocean (Table 1). ODP
Site 1090 on the southern flank of the Agulhas Ridge in the Southern Atlantic Ocean (42°54.8'S, 8°53.9'E, water depth
3,702m), ODP Site 689 on the southern flank of the Maud Rise in the Southern Atlantic Ocean (64°31'S, 3°6'E, water
depth 2,253m) and ODP Site 748 on the southern part of the Kerguelen Plateau in the Southern Indian Ocean
(58°26.45'S, 78°58.89'E, water depth 1,290.9m). All the sites lie on topographic highs. The Agulhas Ridge comprises
an elongate part of the Agulhas-Falkland Fracture Zone (AFFZ). The ridge rises ~ 3000m above the surrounding floor
and constitutes a topographic barrier, having a strong influence on the exchange of water masses (Gruetzner &
Uenzelmann-Neben, 2015) between high and lower latitudes. The Maud Rise is a seamount, its elevation rises almost
3000 m from the seafloor (Brandt et al., 2011). Kerguelen Plateau is a large topographic high in the Indian sector of the
Southern Ocean. We selected samples from the middle Eocene through the E-O boundary, depending on the sample
availability.
Currently, the sites studied are located in the Southern Ocean through the ACC. Site 689 is located well south of the
Polar Front Zone (PFZ),and 748 slightly to the south of the  PFZ and Site 1090 in the Subantarctic zone, between the
Subtropical front (STF) and the Subantarctic Front (SAF) (Figure 1). Across the Eocene-Oligocene transition, the sites
were shallower (Table 1), ranging between ca 1.2 and 3 km paleo water depth. These depths are well suited to capture
signals of export productivity to intermediate-deep waters. Sites 689 and 748 locations were similar to today and site
1090 was as much as 5° farther to the south (Gersonde et al., 1999) (Table 1).
The major lithology from the lower Eocene to the upper Oligocene at the Maud Rise is composed of calcareous and
silicious oozes (Barker et al., 1988). Kerguelen Plateau site is composed mainly of nannofossil ooze and chert (Barron
et al., 1989). Agulhas Ridge is predominantly composed of diatoms and nannofossil ooze, with CaCO$_3$ wt% highly
variable, ranging from non-detectable to 69% of sediment throughout the study interval. (Gersonde et al., 1999) with

rare occurrences and barren intervals of planktic and benthic foraminifera making it difficult to establish stable isotope records at this site.

**Table 1.** Position of the ODP Sites studied in the present-day and in the late-Eocene (~ 37 Ma). Paleocoordinates calculated based on Seton et al. (2012) rotation model.

| Site | Geographic Setting | Latitude | Longitude | Water depth (m) | Paleodepths (m) | Paleo-latitude | Paleo-longitude |
|---|---|---|---|---|---|---|---|
| **1090** | Agulhas Ridge | 42°54.8'S | 8°53.9'E | 3 702 | ca. 3,000-3,300 (Pusz et al. 2011) | ca 47°33'S | ca 1°46.8' E |
| **689** | Maud Rise | 64°31'S | 3°6'E | 2 253 | ca. 1500 (Diester-Haass and Zahn, 1996) | ca 64°19.2'S | ca 2°43.2' E |
| **748** | Kerguelen Plateau | 58°26.45'S | 78°58.89'E | 1 290.9 | ca. 1200 (Wright et al., 2018) | ca 56°48.6'S | ca 75°36' E |

**2.2 Age Models, Linear Sedimentation Rates**

Revised age models for the ODP Site 1090, ODP Site 689 and ODP Site 748 made for this study were based on all magnetostratigraphic and biostratigraphic data available, and both models and data are available fromon the Neptune database-NSB system (Renaudie et al., 2020) (Figures S1-S4). All ages in our study are given in the GPTS standard used by NSB (Gradstein et al. 2012), or have been remapped to this scale from prior studies. Differences between this and more recent GPTS scales in the Cenozoic are minor, and generally less than other age model uncertainties for the sections in our study.

ODP Site 1090 has an age model constructed from shipboard magnetostratigraphic "U-channel" measurements, and the records fit well to the geomagnetic polarity timescale (GPTS) (Channell et al., 2003). Nannofossil biostratigraphy has confirmed the Chron ages (Marino and Flores, 2002), as well as foraminiferal biostratigraphy (Galeotti et al. 2002), strontium isotopes (Channell et al., 2003) and oxygen and carbon isotope data from benthic foraminifera (Zachos et al., 2001; Billups et al., 2002). This integration of several age indicators and their consistency makes this a robust and very well constrained age model.

Magnetostratigraphic data for ODP Site 689 is partially reinterpreted from the measurements originally made by Spiess, 1990. A new high-resolution study of Eocene-Oligocene "U-channel" samples from this site shows a high correlation with the GPTS (Florindo and Roberts, 2005). Calcareous nannofossil datums (Wei and Wise, 1992; Wei, 1992, Persico and Villa, 2002, 2004), planktonic foraminiferal datums (Kennett and Sott, 1990; Thomas, 1990; Berggren et al., 1995) and Argon-argon ($^{40}Ar/^{39}Ar$) dating (Glass et al., 1986; Vonhof et al., 2000) are used to re-calibrate ages for this site.

A high-resolution magnetostratigraphic study from ODP Site 748B was carried out by Roberts et al. 2003 in continuous "U-channel" samples, revising the shipboard analysis from Inokuchi and Heider, 1992. Calcareous nannofossil biostratigraphy (Aubry 1992), planktonic foraminiferal biostratigraphic datums (Berggren et al., 1995), diatom datums (Baldauf and Barron, 1991, Roberts et al., 2003) and strontium isotope ages (Zachos et al., 1999; Roberts et al., 2003) were re-evaluated for a better age model.

Accumulation rate fluxes are obtained by calculating the product of linear sedimentation rates (LSR) and shipboard measured dry bulk densities (DBD), thus a robust age model is crucial for this calculation because it determines the linear sedimentation rates. We use a straightforward LSR calculation between age-depth control points based on magnetostratigraphic data, stable isotopes and biostratigraphic data. Mass accumulation rates (MARs, mol cm$^{-2}$ kyr$^{-1}$) were calculated using LSR based on the above age models multiplied by DBD.

Since bio-Ba AR is a direct function of LSR, it is essential to evaluate any possible biases due to this. Figure S5 shows a comparison of linear sedimentation rates and bio-Ba AR. This comparison showed a high amplitude peak at ODP Site 1090, with LSR of 4.11 cm kyr$^{-1}$ at the late Eocene, this high rate is based on a very constrained model. The LSRs for ODP Site 689 vary from 0.1 cm kyr$^{-1}$ during the early Oligocene to up to 1.3 cm kyr$^{-1}$ in the late Eocene, whereas ODP Site 748 has more uniform values during the late Eocene. The available age data for our sites allow some variation in the placement of the line of correlation, and thus the precise timing and magnitude of sedimentation rate changes on the scale of ± ca 0.5 m.y. are not well constrained. Patterns and calculated values over longer time scales are however thought to be robust.

## 2.3 Stable Isotope Analyses

Stable isotopes of carbon and oxygen were measured both on the bulk fine fraction (<45μm) and benthic foraminifera. Bulk sediments were oven-dried and washed through different sieve sizes (125 and 45μm). Smear slide observations indicate that the main carbonate composition of the fine fraction is coccoliths, therefore stable isotopic compositions of bulk fine fraction (<45 μm) reflect primarily nannofossil isotope signals. Contamination by non-coccolith carbonate such as fragments of foraminifera shells is minimal (Figure S5). Fifteen to twenty tests of benthic foraminifera (*Cibicidoides* spp.) were picked from the >125-μm-size fraction. Foraminiferal tests were ultrasonically cleaned using ethanol and oven-dried. Stable isotopic analyses were carried out at the Stable Isotope Laboratory of the Museum für Naturkunde (Berlin, Germany) on a Thermo Isotope Ratio Mass Spectrometer. All values are reported in the δ-notation in parts per mil (‰) relative to the Vienna Pee Dee Belemnite (VPDB). In this study, we applied an adjustment of +0.64‰ (Shackleton and Opdyke, 1973; Shackleton et al., 1984) to all $\delta^{18}O$ values of the benthic foraminifera *Cibicidoides* to account for disequilibrium effects.

## 2.4 Barium Analyses and Biogenic Barium as a Paleoproductivity Proxy

Barium (Ba) and aluminium (Al) were analysed by ICP OES, performed at the ElMiE Lab at the German Centre for Geosciences (GFZ, Potsdam, Germany) using a 5110 spectrometer (Agilent, USA). The analytical precision and repeatability were generally better than 2% and it is regularly tested by certified reference material and in-house standards. For preparation, 2g of each sample w grounded to assure grain size distribution, and digested by $Na_2O_2$ fusion and HCl using ultrapure reagents, following the method by Bokhari and Meisel (2017). Intensity calibration was performed by external calibration using the same batch of solvent to ensure matrix matching. The analytical blank was negligible compared to the sample concentration.

Using barium as a paleoproductivity proxy requires some adjustments because other biogenic sources may contribute to the barium content in the sediment. Detrital aluminosilicate may affect the barium signal in Southern Ocean sediments. In order to solve this issue and reveal aluminosilicate contributions, the Biogenic Barium calculation was used as proposed by Dymond et al., 1992, following Eq (1):

Biogenic Barium (Bio Ba) = (Ba total) $_{sample}$ - (Ba/Al) $_{bulk\ continental\ crust}$ x Al $_{sample}$ (1)
This assumes that the aluminium (Al) concentration and the average continental crust abundance are representative of
the detrital Ba component. The Ba/Al crust ratio of 0.0075 is the global average value from sedimentary rocks as
suggested by Dymond et al. (1992). This value is based on various compilations of elemental abundances in crustal
rocks. This normative calculation potentially introduces uncertainty in samples with high and variable detrital barium,
but considering that clay assemblages and weathering regimes were relatively constant during the early Paleogene in the
Southern Ocean, therefore, the crustal ratio probably did not vary much (Robert et al. 2002).
**2.5 Data Compilation**
**2.5.1 Neodymium Isotope Data**
Neodymium isotopes in seawater reflect the different weathering sources of neodymium that affect each water mass.
The isotope values act as conservative elements during ocean mixing. They are therefore a robust water mass tracer, and
are faithfully archived in sediments (Piepgras and Wasserburg, 1982; Martin and Haley, 2000). Their behaviour in
seawater and the conservation of the signal in sediments make them a valuable proxy for paleoceanographic studies and
past ocean circulation reconstruction. Fossilised fish teeth are commonly used and considered robust archives to extract
Nd isotopic signatures because they incorporate and preserve their Nd signature during very early diagenesis (Martin
and Scher, 2004), and they can be found in deep-sea sediment samples all over the world and in many geologic time
intervals. The Nd signal is given in εNd, where εNd is the ratio $^{143}Nd/^{144}Nd$ of a sample relative to the same of the bulk
Earth, in parts per 10,000.
In this study, we compiled published Nd isotope data from fossil fish teeth, from the same Ocean Drilling Program
(ODP) sites that we investigated in the Southern Ocean (ODP Site 1090 Agulhas Ridge, Scher and Martin, 2006; ODP
Site 689 Maud Rise, Scher and Martin, 2004; and ODP Sites 738, 744 and 748 on the Kerguelen Plateau, Scher and
Delaney, 2010; Scher et al., 2011; Scher et al., 2014; Wright et al., 2018) and explore the Nd isotope variability to
examine the intrusion of waters from the Pacific to the Atlantic sector of the Southern Ocean. We then used these data
and our records to explore the evolution of the Southern Ocean circulation and significant circulation changes across the
Eocene-Oligocene transition. Sources of Nd isotope data are given in Table S1.
**2.5.2 $p$CO$_2$ Data**
A variety of geological proxies have been applied in numerous studies to reconstruct the partial pressure of atmospheric
CO$_2$ ($p$CO$_2$) during the Cenozoic Era (e.g. Pagani et al., 2005, 2011; Künschner et al., 2008; Retallack et al., 2009;
Beerling & Royer 2011, Anagnostou et al., 2016, 2020). Given the low published sampling density through the critical
Eocene-Oligocene interval, we compiled published $p$CO$_2$ data from marine and terrestrial proxies that have been
identified as reliable for reconstructing $p$CO$_2$ during this period. The marine geochemical proxies include alkenone-
based estimations, carbon and boron isotope ($\delta^{11}$B) composition of well-preserved planktonic foraminifera calcite.
Proxies from the terrestrial reservoir include Paleosols and Stomatal frequencies. Our atmospheric CO$_2$ compilation
(Table S2) consists to our knowledge all the currently available proxy data on Eocene and Oligocene $p$CO$_2$ records,
including all data compiled in previous syntheses through extensive scientific community efforts in paleo CO$_2$ database
(The Cenozoic CO$_2$ Proxy Integration Project Consortium, 2023), such compilations are commonly used to estimate

past $p$CO$_2$, although it is known that there are limitations and variation among them (IPCC, 2021). Zhang et al. (2013) specifically argued that compositing limited, short time interval data from different proxies, and different localities is likely to introduce significant short-term bias at individual data series end-points into the resulting fitted curve, and instead generated a 40 My long history of Cenozoic $p$CO$_2$ using a single proxy from a single section (Site 925 in the equatorial Atlantic). We consider this study's results to be the best and most complete single source of information on the Cenozoic trend of atmospheric $p$CO$_2$. However, precisely because of the substantial amounts of between proxies and between locality variation, data using a single proxy and from a single site is also potentially not representative of global $p$CO$_2$ history. We thus use both the single site results of Zhang et al. (2013) and the full, multi-site and multi-proxy compilation (Supplementary material) in evaluating our own study's results.

**3 Results**

**3.1 Biogenic Barium**

Biogenic Barium accumulation rate (bio-Ba AR) records (Figure 2) show a pronounced rise in the late Eocene when the values were up to twice as high as in previous periods for all sites studied. At Kerguelen Plateau ODP Site 748, we also observe a previous and smaller increase around the middle Eocene Climatic Optimum (MECO, ca 40 Ma). At Maud Rise the increase began at ca 38.3 Ma and persisted for around 1.5 Myr. bio-Ba ARs show a high value in the Agulhas Ridge at ca 36.8 Ma which is induced by a high sedimentation rate (Figure S5a). Although export productivity was higher (maximum values to about 16.8 µmol bio-Ba cm$^{-2}$ ky$^{-1}$) at Maud Rise compared to the other sites -the Kerguelen Plateau reached maximum values of about 14.3 µmol cm$^{-2}$ ky$^{-1}$ and the Agulhas Ridge site 13.74 µmol cm$^{-2}$ ky$^{-1}$, the records show a high degree of temporal correspondence in the late Eocene peak (ca 36.8 Ma). Bio-Ba values were low at all sites between ca 36 and 34.5 Ma. Between ca 34.5 Ma and ca 33.3 Ma, which includes the EOT interval, bio-Ba AR increased in both sites of the Atlantic Sector, but these increases were not very concurrent between the sites investigated. On the Agulhas Ridge, ODP Site 1090, the rise in bio-Ba (from 7.37 to 20.46 µmol cm-2 kyr$^{-1}$) is observed in the very latest Eocene (ca 34.3), just before the Oi-1 event. At Maud Rise, ODP Site 689, the increase is not observed until ca 1 Myr after, in the early Oligocene (maximum value 16.25 µmol cm$^{-2}$ kyr$^{-1}$ at ca 33.3 Ma). On the Kerguelen Plateau, ODP Site 748, the increase in export productivity registered by bio-Ba during the Oligocene is notably smaller than in the Atlantic sites, with values not higher than the low values observed during the Eocene.

Our bio-Ba results are in general concordant with the temporally more limited data obtained by prior studies of Southern Ocean sites (Anderson and Delaney, 2005, Site 1090; Diester-Haass and Faul 2019, Site 689) (Figure 2). However, our results for Kerguelen Plateau Site 748 differ from those of Faul and Delaney, 2010 for nearby Site 738, where the latter estimate bio-Ba accumulation rates up to twice those obtained in our study of Site 748. The differences may be due to the different locations of the two sites, Site 738 is located several degrees further south, and in ca 1 km deeper water depth. The bio-Ba proxy is also very sensitive to sedimentation rates, and the differences may be due to the poor age control for Site 738 which in the studied time interval consists only of a few rather scattered biostratigraphic events (Figure S4a), resulting in substantially different age models between our study, Faul and Delaney (2010), and other recent studies of this site, e.g. Huber and Quillevere (2005). In these studies the location and extent of hiatuses, and the uniformity of sedimentation rates varies considerably (SOM Figure S4b). The age model for Site 748 by contrast (Figure S3) is very well constrained by coherent biostratigraphic events from multiple groups of

microfossils, Sr isotope stratigraphy and paleomagnetic stratigraphy, and we therefore accept the results from Site 748
as being more reliable.
When the data for individual sites is composited together, the behaviour of the Southern Ocean region can be roughly
estimated, even though our geographic coverage (lacking data from the Pacific/New Zealand sector) is incomplete and
thus may not be entirely representative of the Southern Ocean as a whole. A lowess curve fit to the composited bio-Ba
data shows that the key patterns noted in individual records are retained in the composite signal, and thus that Southern
Ocean productivity can be characterised as having had two intervals of high values at around 37 and 34 Ma.
**3.2 Oxygen and Carbon Isotopes**
Our new oxygen (Figure 3C and E) and carbon (Figure 3D and F) stable isotope data allow us to identify previously
noted trends and distinct events during the period studied. Benthic $\delta^{18}O$ values exhibit a consistent increasing trend
during the late Eocene indicating the overall decrease of oceanic bottom water temperatures. A sharp increase occurs at
the Eocene-Oligocene transition (between 33.9 and 33.3 Ma) at both sites examined. This rapid shift has been observed
in several sites in the Southern Ocean (e.g., Muza et al., 1983; Miller et al., 1987; Mackensen and Ehrmann, 1992;
Zachos et al., 1996; Billups et al., 2002; Pusz et al., 2011) and it is well established as a global signal (Zachos et al.,
2001). It is generally interpreted as a combination of deep ocean water cooling and major ice growth on the Antarctic
continent (Zachos et al., 2001). At Site 689, the planktic $\delta^{18}O$ curve almost mimics the benthic one. The $\delta^{18}O$ values
measured on fine fraction reveal a heavier trend more pronounced at ODP Site 748 (Kerguelen Plateau) compared to
ODP Site 689 (Maud Rise). During the late Eocene, around 37 Ma, heavier $\delta^{18}O$ values are observed in both the Atlantic
and Indian Sectors of the SO, the benthic/fine fraction ratio declines, indicating more homogenous temperature in the
water column. Both benthic and planktic foraminifera $\delta^{13}C$ records show fluctuations across the period studied, with
low values across the Eocene-Oligocene boundary, followed by an increase that accompanied the $\delta^{18}O$ increase and low
values again in the upper Oligocene. The benthic trend is also observed by previous data from the same sites
(Mackensen and Ehrmann, 1992; Diester-Haass and Zahn, 1996; Bohaty et al., 2003). The fine fraction records show
elevated $\delta^{13}C$ values between Late Eocene to Early Oligocene, followed by a decreasing trend during the Oligocene
(from ca 33.2Ma). At Site 748, the fine fraction $\delta^{13}C$ curve shows less fluctuation than the benthic curves during the
middle Late Eocene. A synchronous $\delta^{13}C$ increase (ca 0.6‰ shift) is observed at 36.5 Ma. Elevated fine fraction $\delta^{13}C$
values are observed from the late Eocene until the early Oligocene, coherent with previous studies (Bohaty et al., 2003),
while the benthic values stay low during the same period (Figure 3D and F).
**3.3 $p$CO$_2$ Proxies**
As noted above, given the complexities and potential biases of compiling data from different proxies and different time
intervals, we prefer to use the single site single proxy time series of $p$CO$_2$ from Zhang et al. (2013). This data (Figure 4)
shows two peaks in the late Eocene, with a maximum for the entire study interval at ca 37 Ma and a smaller peak at ca
34.5 Ma, and a rapid drop of over 200 ppm from nearly 1000 to ca 750 ppm in the earliest Oligocene (ca 33.5 Ma).
Despite the limitations of multi-proxy, multi-site compilations, the compiled data (Table S2; Figure S7) shows the same
basic features, nor does the result appear to be sensitive to the precise choice of data to include in the analysis.
**4 Discussion**

### 4.1 Correlation of Productivity to Ocean Circulation, Glaciation and $p$CO$_2$ Change

Intervals of high export productivity (bio-Ba) exhibit synchronous changes with intervals of changes in oxygen and carbon stable isotopes and $p$CO$_2$ data (Figures 3 and 4). Notably, these intervals occur in the late-Eocene ($\sim$ 37 Ma) and at the E/O boundary ($\sim$34 Ma). This correlation is highly suggestive that ocean circulation, glaciation, productivity and atmospheric $p$CO$_2$ changes are interconnected. It could be assumed that (1) primary processes, such as ocean circulation and glaciation are independently driving productivity and $p$CO$_2$ changes, or (2) a cascading effect, with glaciation and ocean circulation influencing productivity and, consequently, altering atmospheric $p$CO$_2$ levels. While we cannot in our study distinguish between these two possibilities, we explore the second, as proposed by Egan et al. (2013), in which changes in ocean conditions due to tectonic-climate drivers affect productivity, and the latter in turn $p$CO$_2$ levels.

### 4.2 Late Eocene Productivity Event and its Potential Impact

The noticeable bio-Ba AR peak at $\sim$ 36.8 Ma (Figure 2), suggests an important, ca 1 My long event of approximate doubling of export productivity during the late Eocene, preceding the significant cooling and the first formation of large Antarctic ice sheets at the Eocene-Oligocene boundary. The temporal synchronicity among different site locations in the Southern Ocean suggests that the process driving this enhanced export productivity in the late Eocene occurred throughout the Southern Ocean, requiring a mechanism that increased the delivery of nutrients to the surface ocean. Our findings corroborate previous paleoproductivity studies that indicate an increase in export productivity in the Atlantic Sector of the Southern Ocean during this time period. Anderson and Delaney (2005) found several peaks in productivity indicators at the Agulhas Ridge during the same time interval, and benthic foraminiferal accumulation rates show an increase in paleo-primary productivity on Maud Rise (Diester-Haass & Faul, 2019) (Figure 2). A pronounced opal abundance peak is also documented by Diekman et al. (2004) between 37.5 and 33.5 Ma at the ODP Site 1090. Our results now show that the 37 Ma event extended at least as far as the Kerguelen Plateau in the Indian Ocean sector, with a substantial peak around 37 Ma and an earlier one near 40 Ma, thus affecting a large portion of the Southern Ocean region.

A direct analysis of the impact of the potential significance of this event for the development of late Eocene global climate depends on two key factors: the extent to which the increased productivity contributed to enhanced carbon sequestration, and the magnitude of sequestration over the ca 1 my interval of enhancement. Understanding the impact of productivity on carbon sequestration for the Eocene oceans however is complicated by the lack of knowledge of several factors that influence this process. These factors include our ability to estimate the impact of higher productivity on carbon sequestration is limited, as many of the factors that affect this in the modern ocean are poorly understood for Eocene oceans (export efficiency to the subsurface waters, rates of transport and degradation in the water column and upper sediment layers, organic carbon content of Southern Ocean Eocene pelagic sediments; as well as transport of organic carbon by subsurface water layers in the late Eocene oceans to lower latitude areas of productivity and sequestration. We therefore cannot directly calculate the impact that the observed productivity change had on pCO$_2$. Instead we take an indirect approach, comparing the productivity history to the history of pCO$_2$ and potential drivers of productivity change such as ocean circulation and climate change.

### 4.3 Surface Water Changes in Physical Conditions in the Late Eocene

The late Eocene is generally accepted as a time interval of gradual cooling of Southern Ocean waters. Indeed,
biomarker-based temperature estimates reveal substantial (3-5 °C) high latitude sea surface temperatures (SST) cooling
within the late Eocene (Liu et al., 2009; O'Brien et al., 2020). Our fine fraction stable oxygen isotopes confirm this
cooling trend following MECO, with a distinct peak at 37 Ma during the Eocene, matching the peak cooling reported by
O'Brien et al. (2020). This interval of maximum $\delta^{18}O$ values occurred during the same interval in which export
productivity increased (Figure 2). In this interval the difference in the $\delta^{18}O$ gradient between benthic foraminifera and
fine fraction (nannofossil) carbonate is less pronounced. This increase in similarity is interpreted as having been caused
by a decrease in water column stratification and enhanced vertical mixing.
This change in export productivity in the late Eocene is coeval with a change towards increasing variability of carbon
stable isotopes ($\delta^{13}C$) of benthic foraminifera (Figure 4). Although $\delta^{13}C$ in individual sections can also represent local
effects, usually there is a strong component of global changes in carbon reservoirs, and indeed our local measurements
closely align with the global curve (Figure 3B, D and F). In the global context, a shift towards more positive values in
the benthic $\delta^{13}C$ at ~ 37 Ma indicates a carbon cycle perturbation. This shift coincides with the export productivity
changes observed in our study. One possible explanation is that higher productivity may have elevated the export flux of
organic carbon to sediments, thereby increasing the marine organic carbon burial and preferentially scavenging the
lighter 12C from the carbon pool.
**4.4 Oceanographic Circulation Drivers of the Late Eocene Productivity Change**
We propose that the main cause for the productivity increase observed in the late Eocene is the upwelling of nutrient-
rich deep waters. Understanding however the physical oceanographic mechanisms that led to increased upwelling
throughout the Southern Ocean requires examining links between the different processes that occurred at that time
period. Changes in paleoceanography during the Paleogene were significantly mediated through tectonic re-
organisation, such as the Southern Ocean gateways opening (i.e., the Drake Passage and the Tasman Gateway), changes
in the Atlantic-Arctic gateway and in the Tethys Seaway.
In this context, the Southern Ocean circulation during this time period is still debated due to uncertainties concerning
the opening of the gateways that led to the development of the Antarctic Circumpolar Current (ACC). Estimates for the
onset of the modern-like ACC have not reached a consensus yet and vary from as early as middle Eocene (ca 41 Ma,
Scher & Martin, 2006; ca 35.5 Ma, Stickley et al. 2004) to middle Oligocene (ca 23Ma; Pfuhl and McCave, 2005). This
inconsistency suggests that the onset of ACC could have been a gradual or an intermittent change. Further, local proxy
records cannot distinguish between regionally developed fronts and true circumpolar flow (i.e. ACC).
In addition to $\delta^{18}O$ and $\delta^{13}C$, $\epsilon Nd$ has been used to identify circulation changes and water masses exchange through the
Eocene (Scher and Martin, 2004, 2008; Scher et al., 2014; Huck et al., 2017; Wright et al., 2018). Nd isotopes are one of
the most robust tracers of water mass origin (Frank, 2002). The residence time of Nd in oceans is much shorter (300-
1000 years) than ocean mixing time and is thus distinct at a given location. Further, the isotope composition of the Nd
ocean budget is solely determined by terrigenous contribution. The latter is balanced by Nd sinks that remove Nd
quantitatively, yet this only influences the net budget and thus the magnitude and/or swiftness of changes to the $\epsilon Nd$
composition. However, mixing of water masses, e.g. through lateral or vertical mixing, can also cause changes as long
as they occur more rapidly than the residence times.
In the late Eocene, starting at 37 Ma, Scher and Martin (2004) found a dramatic positive shift in ε-Nd values in the
Atlantic sector of the SO that they interpreted as the influx of Pacific deep waters, due to its characteristic of more
radiogenic (positive) waters, not previously observed in the Atlantic Ocean. Recently published Nd isotope records from
the Kerguelen Plateau (Wright et al., 2018) revealed a long-term negative trend during the late Eocene, which also
suggests that the water mass mixing between the Pacific and Atlantic preceded 36 Ma. εNd (t) records from the
Kerguelen Plateau in fact showed values comparable to modern CDW during the Oligocene, inferring water mass
composition similar to the present day. Thus, Nd isotope data support at least partial opening of Drake Passage by the
late Eocene (before 36 Ma), consistent with plate tectonic reconstructions (Livermore et al., 2005, 2007). Regardless of
the depth, Neodymium isotope evidence for late Eocene opening of the Drake Passage suggests that increased fetch for
surface flow and changing deep water composition could have had changes in the surface water conditions in the South
Atlantic sector of the late Eocene Southern Ocean.
This has been explicitly demonstrated in a recent modelling study conducted by Toumoulin et al. (2020). They
demonstrate that the Drake Passage opening, even at shallow depth, notably connects prior regional frontal systems
together, thereby allowing the formation of a proto-ACC; and has a strong effect on the Southern Ocean Eocene water
mass structure, inducing ocean cooling in most of the Southern Hemisphere. These temperature changes are not linear
and differ from one region to another, with DP opening causing changes in the mixed layer depths and provoking
different responses in the Atlantic and Indian Sectors of the SO. In the Atlantic and Indian Ocean sectors in particular,
very deep seasonal mixing (several hundred meters) over broad areas of the entire region is replaced by more moderate
levels of mixing (generally ca 200 m or less), except near the proto-polar front region, where seasonal mixing of 300-
400 m still occurs. Vila et al. 2014 have found nannofossil assemblages characteristic of cool sea surface waters in the
late-Eocene in Kerguelen Plateau samples. Cooler temperatures are coeval with the paleoceanographic re-organization
and intensified upwelling that we infer for this time period, while differences in the depth of the mixed layers between
ocean basins may explain the different magnitude of export productivity observed in the Atlantic and Indian sectors of
the SO.
The wind-driven eastward flow and the characteristic fronts of the modern ACC support the upwelling of nutrient-rich
water to the surface and consequently high levels of productivity. On the balance of evidence, it seems that the export
productivity seen in our data in the late Eocene is likely to have occurred in response to a proto-ACC front's
development and its associated upwelling. The inferred onset of a proto-ACC in the late Eocene and our finding of
increased upwelling fits the hypothesis that ACC type circulation itself helps drive the AMOC circulation (Toggweiler
and Bjornsson, 2000; Katz et al., 2011; Sarkar et al., 2019). A proto-ACC causes SO upwelling, and thus provides
support for increasing AMOC-like circulation in the late Eocene as an additional cause of increased upwelling as a
causative mechanism of the export productivity event. Temperature asymmetry between Northern and Southern
Hemisphere and comparisons between benthic $\delta^{13}C$ records provide evidence for the strengthening of the AMOC in the
late Eocene (Elsworth et al., 2017).

**4.5 Eocene-Oligocene Boundary Productivity Changes**

The earliest Oligocene, following the E/O boundary, has been suggested as a period of a significant rise in biological
productivity in high southern latitudes (Diester-Haass, 1995, 1996; Diester-Haass and Zahn, 1996, 2001). However, in
contrast to the late Eocene event, the export productivity changes across the Eocene-Oligocene boundary observed in

our study were not always concurrent between the sites investigated (Figure 2). In the Atlantic sites, export productivity increases and decreases several times from the late Eocene to early Oligocene. We thus argue that the fluctuations in export productivity that occurred in the Southern Ocean during this global climatic re-organization are more strongly modulated by local parameters, whereas the late Eocene productivity event is more uniform and reflects the global re-organization of ocean circulation. If the trends observed in export productivity across the EOT were regulated only by global, or at least regional temperature and circulation changes, then we would observe significant changes also in the Indian sector of the Southern Ocean. It seems however that productivity increase was more pronounced in the Atlantic sector of the Southern Ocean.

Today, the Southern Ocean (SO) has a frontal system that strongly impacts circulation, primary productivity and the entire climate system (Chapman et al., 2020). The Antarctic Polar Front (APF) is particularly important for controlling nutrient distribution. Latitudinal variations of the APF for example have been shown to alter regional productivity over the glacial cycles (Kim et al., 2014, Thole et al., 2019). The causes of the lack of significant export productivity changes in the Indian sector of the Southern Ocean during the early Oligocene after the Eocene-Oligocene boundary are unclear. The Kerguelen Plateau may not have been located in a position favourable to nutrient-rich upwelling. In addition, the regional frontal migration may have been more intense in the Atlantic sector compared to the Indian sector of the SO.

**4.6 A Scenario for Southern Ocean Productivity and Circulation Change in the Late Eocene**

The patterns of productivity change seen in our study can be placed in an (admittedly speculative) scenario, which is at least compatible with prior studies and modelling of conditions in the late Eocene austral ocean region and Antarctica. In the earliest interval covered in our study (ca 40-38 Ma) productivity was in most sites fairly low (Figure 5a). At this time there is little evidence for a significant influx of Pacific waters into the Atlantic, and the Drake Passage is thus assumed to be effectively closed to ocean circulation.

During the 38-36 Ma interval, evidence summarised by Scher et al. (2014) suggests that a significant, if transient, glaciation event occurred on the Antarctic continent-the Priabonian oxygen maximum, or PriOM. If sufficient in magnitude this would have significantly affected circulation throughout the austral ocean region, with strengthened temperature gradients, invigorated circulation (Houben et al., 2019), and increased upwelling (Goldner et al, 2014). This would account both for the substantial increase in productivity, and the broad geographic extent of the increase seen in our data (Figure 5b). The cause of this glaciation event is unknown, but may be related in part to the trend in the late Eocene towards lower atmospheric $p$CO$_2$ interacting with orbital fluctuations in polar insulation as explored in model simulations by Van Breedam et al. (2022).

With the end of transient glaciation, the atmospheric forcing of ocean circulation would have declined, and with it the high levels of productivity seen in our data (Figure 5c). However, by this time (ca 36-34 Ma) the Nd isotope data suggests that a significant influx of Pacific water was reaching the South Atlantic sector of the austral ocean (Scher and Martin, 2004), and consequently, the Drake Passage must have been at least partially open. This would have resulted in, if not as strong as during the PriOM, nonetheless stronger circumpolar circulation in a proto-ACC, increased upwelling, and increased nutrient availability from Pacific-sourced deep waters. The locus of high productivity would have become however more cantered near the proto-ACC, which at that time, according to the model results of Toumoulin et al. (2020) was located a few degrees north of the current location of the ACC. The high productivity and accumulation of biogenic opal seen at Site 1090, fortuitously located at this time in this region can be thus be explained, as can the lower

relative productivity of Site 689, located much further to the south and thus outside the region primarily influenced by
the proto-ACC system.
Lastly, at the E/O boundary itself (Figure 5d), the well known major shifts in oxygen isotopes signal the formation of a
full continental ice-sheet, which would have in turn driven a renewed increase in circumpolar ocean circulation- a full,
if early form of the ACC, and dramatically increased levels of productivity, again however primarily near the ACC
region.

### 4.7 Possible Implications to the EOT Global Cooling

Our $p$CO$_2$ compilation shows that carbon dioxide levels declined gradually from ca 1200 ppm in the late Eocene to ca
750 ppm across the EOT (Figure 4 and Figure S7). There are different processes involved in the oceanic uptake of CO$_2$.
The solubility pump is a physical-chemical process that promotes gas transfer between the atmosphere and seawater in
order to achieve chemical equilibrium. This process depends on temperature, in which the solubility increases as
temperature decreases. Evidence for cooling of surface waters observed in the Southern Ocean (Liu et al., 2009;
Hutchinson et al., 2021) could have favoured the ability to dissolve atmospheric CO$_2$, contributing to the drawdown of
$p$CO$_2$ during the late-Eocene.
Silicate weathering has been suggested to play an essential role in regulating CO$_2$ across the EOT (Zachos and Kump,
2005). On geologic time scales, chemical silicate weathering is considered to modulate atmospheric CO$_2$ levels through
a negative feedback mechanism (Berner et al., 1983). Weathering of silicate rocks provides alkalinity to the oceans,
acting as a sink for atmospheric CO$_2$, thereby influencing global climate. In addition, increasing silicate weathering
enhances primary productivity through the delivery of nutrients to the ocean. Intensified weathering is supported by Os
isotope records, showing an anomaly before the EOT, at ca 35.5Ma (Dalai et al., 2006).
High export productivity potentially modulates CO$_2$ by two major mechanisms, operating over very different time
scales. Productivity maintains a gradient in dissolved CO$_2$ between the surface and deep ocean by exporting organic
carbon from the surface into deep ocean waters, This in turn lowers the concentration of CO$_2$ in surface waters, causing
more atmospheric CO$_2$ to be drawn out of the atmosphere into dissolved surface ocean CO$_2$. It is estimated that this
mechanism causes $p$CO$_2$ to be approximately 200 ppm lower than it would in a purely abiotic ocean (Volk and Hoffert
1985). In the second mechanism, in areas of the ocean with high export productivity a (generally rather small) fraction
of biologically captured carbon (both soft tissue and carbonate from coccolithophores and planktic foraminifera)
escapes remineralization in the water column and is buried in ocean bottom sediments, where it is sequestered for
(typically) many millions of years. Changes in either mechanism can contribute to the decline of atmospheric carbon
dioxide, thereby intensifying the cooling trend. Therefore, the observed increase in exportproductivity in the late-
Eocene over 1 My in multiple sites in the SO, and the temporal correlation with the changes in $p$CO$_2$ proxy records
showed in our study is compatible with the hypothesis proposed by Egan et al., 2013.This suggests that the heightened
export productivity identifying in our study is a potential candidate that may have provided important positive feedback
to the $p$CO$_2$ decline.
However, this correlation is suggestive rather than conclusive, given the complexities of the carbon cycle. For instance,
variations in the efficiency of the carbon pump, remineralization (Griffith et al., 2021), the relationship between
nutrients availability and plankton utilization and the dynamics of shelf-deep sea carbonate (Sluijs et al., 2013) can
significantly influence the relationship between export productivity and $p$CO$_2$ levels and adds complexity to our

understanding. Lmited data for the late-Eocene on some critical parameters to our understanding (e.g. sedimentary $C_{org,}$ Olivarez Lyle & Lyle, 2006) further contributes to the uncertainties in establishing a causal link. Moreover, it is essential to note that upwelling, while promoting nutrient-rich conditions favourable to productivity and the long-term sequestration mechanism, also contribute to outgassing of $CO_2$ into atmosphere, influencing the carbon balance in the short-term mechanism. This adds another layer of complexity to the carbon cycle dynamics and its relationship with productivity changes. While our findings propose a potential link, it is crucial to recognize that this alone does not constitute proof of the export productivity directly influenced late Eocene $pCO_2$ levels.

Despite some modelling studies showing that circulation changes were not the main factor in driving the cooling and glaciation on Antarctica (e.g. Huber et al., 2004; Huber and Not 2006; Sijp et al., 2011), circulation changes still had a significant impact, with e.g. Southern Ocean sea surface temperature declines of 2-4°C (Toumoulin et al., 2020), affecting the atmosphere-ocean $CO_2$ equilibrium. A succession of events may have contributed to the evolution of climate: thermal isolation of Antarctica, glacier formation, increasing intensity of silicate weathering, together with upwelling of nutrient rich and cold deep waters, leading to high biological productivity, then declining $pCO_2$. Moreover, because the Atlantic Meridional Overturning Circulation (AMOC) affects the distribution of tracers such as temperature, dissolved inorganic carbon (DIC), alkalinity and nutrients (Boot et al., 2021), the strengthening of AMOC could explain the further decrease in atmospheric $CO_2$ via biological export productivity. Elsworth et al., (2017), for example, suggest that enhanced weathering is driven by intensified AMOC in the latest Eocene due to increasing AMOC causing differential global distribution and increase of surface temperatures and precipitation over land areas. These factors together suggest that E-O changes in AMOC also may play an important role as a driver of $CO_2$ decline.

Taken together, the above processes indicate that a variety of positive feedback contributed to Antarctic glaciation from about 37 Ma onwards. Recent evidence suggests that continental-scale Antarctic glaciation initiated in the late Eocene (Scher et al., 2014; Carter et al., 2017). Our results indicate that significant changes in Southern Ocean export productivity preceded the E/O boundary by approximately 3 million years. These trends are likely to be a response to the combination of the intensified processes that had been in place since the late Eocene. The correlation between our Southern Ocean productivity peaks and decline in global pCO2 records suggest that biological productivity may have played an important role in the drawdown of $pCO_2$ levels. Specifically enhanced $CO_2$ fixation by phytoplankton and carbon sequestration in seafloor sediments increased via an increase in the biological pump may have contributed to decrease atmospheric $CO_2$, and through a positive feedback from declining Southern Ocean surface water temperature enhanced boosting the cooling trend. The establishment of Antarctic glaciation may thus have been influenced significantly by enhanced productivity.

**4.8 Limitations and Future Directions**

Our study has numerous limitations. Our data on paleoproductivity does not cover the full time interval in all of the sites studied, and our geographic coverage is still incomplete. In particular, we have not examined sections from the Pacific sector, or the influence of the Tasman gateway. Most of our interpretations are based on a single productivity proxy - biogenic Barium. While this proxy is well established and gives coherent results in our study, productivity proxies are known to have complex behaviours, and results using different proxies might be at least somewhat different. Our suggestion that elevated late Eocene Southern Ocean productivity might have affected global carbon sequestration is only a speculation based on general characteristics of the ocean carbon system, and much more detailed study of both

actual sequestration values in sediment, and the impact on atmospheric $p$CO$_2$ are still needed. Our interpretative
scenario attributing productivity changes to a combination of Drake Passage opening and continental scale glaciation on
Antarctica are also purely qualitative and need further study. Despite these limitations, our study sheds new light on the
late Eocene oceanic precursors of the Eocene-Oligocene glaciation event-the most dramatic climate change of the
Cenozoic.

**5 Conclusions**

Our bio-Ba data provide important records of the Southern Ocean productivity history across the EOT. These data show
that export productivity increased significantly in the late Eocene in the Southern Ocean and was affected by ocean
circulation changes. The development of a regionally varying circumpolar polar flow (proto-ACC) and the associated
frontal system proposed by other recent studies is likely to have contributed to the enhanced productivity in the
Southern Ocean through the intensification of upwelling (H1).
Our results show that increasing Southern Ocean productivity in the late Eocene to earliest Oligocene is correlated to
global changes in atmospheric $p$CO$_2$ and carbon isotope proxies for organic carbon extraction. This finding points
toward a potential positive climate system feedback, involving ocean circulation changes, enhanced export productivity
and drawdown of atmospheric CO$_2$. Although many studies of the Eocene-Oligocene climate transition try to identify a
single dominant mechanism (e.g. ocean gateway opening vs $p$CO$_2$ decline) for causing the initiation of Antarctica
glaciation, each mechanism plays a different role and has associated complex feedbacks. Our study points toward a
climate feedback system involving ocean circulation, thermal isolation and biological productivity, where several
mechanisms are interconnected and cannot be considered separately. Openings of gateways led to the development of a
circumpolar flow, promoting cooling and increased upwelling that contributed to enhanced ocean carbon pump activity
and the decline of atmospheric carbon dioxide.

**Data Availability**

The supplementary information related to this article is available in the Supplement, the raw data will be available upon
publication in an open-access database (PANGAEA: https://www.pangaea.de).

**Author Contributions**

The manuscript was designed and written by GRF in collaboration with DL. DL updated age models. GRF prepared all
the samples for geochemical analyses. JS ran the barium and aluminium analyses. US generated carbon and oxygen
stable isotope data. $p$CO$_2$ data and neodymium isotopes data were compiled by GRF. Biogenic barium was calculated
by GRF. All authors contributed to editing the manuscript.

**Competing Interests**

The authors declare that they have no conflict of interest.

**Acknowledgments**

This study was funded by the Federal Ministry of Education and Research (BMBF) under the "Make our Planet Great
Again, German Research Initiative", grant number 57429681, implemented by the German Academic Exchange Service
(DAAD).

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

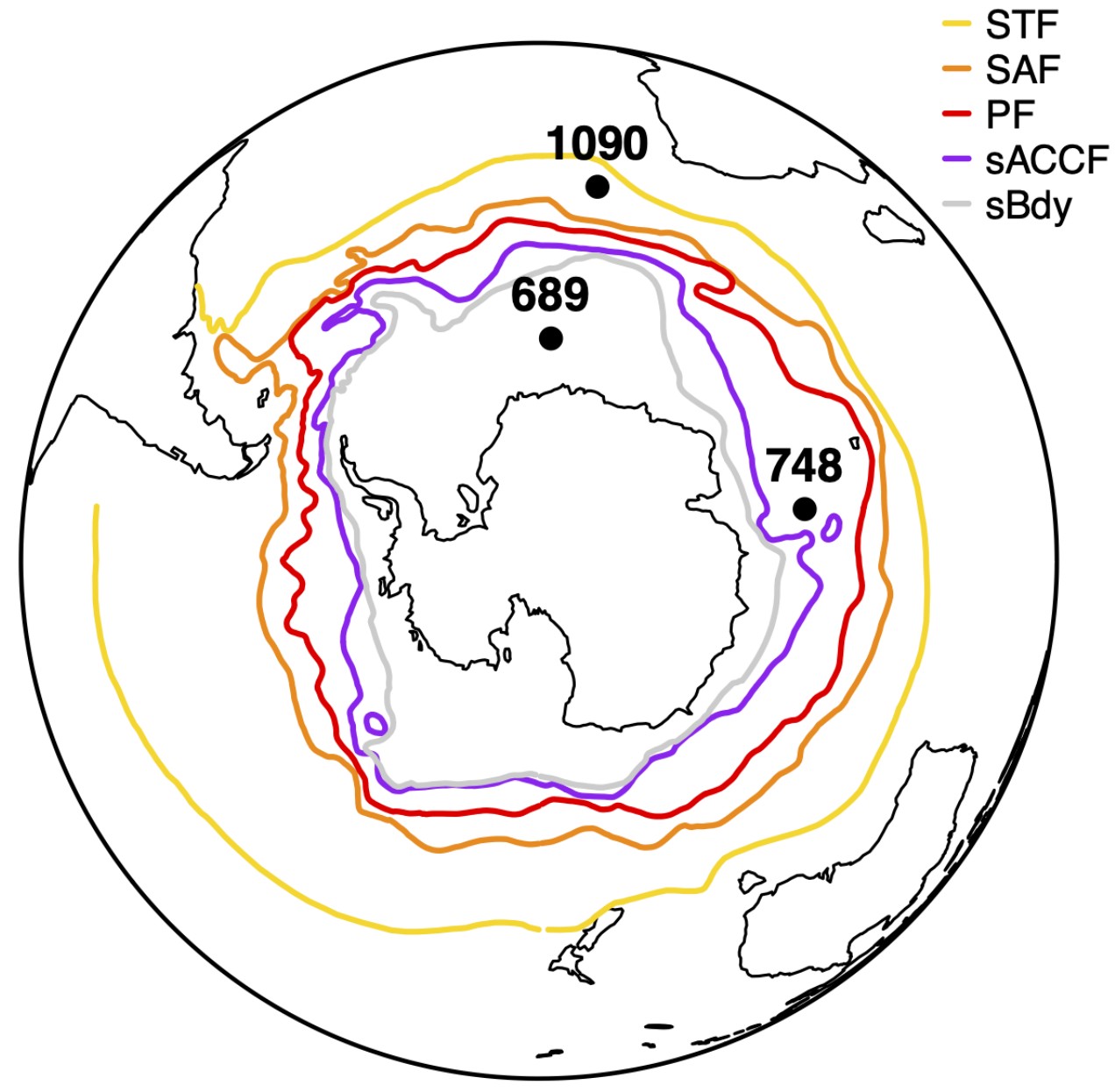

**Figure 1: Schematic Antarctic Circumpolar Current (ACC) and Southern Ocean fronts as determined by Orsi et al., 1995, named from north to south, STF: Subtropical front, SAF: Subantarctic Front; PF: Polar Front and SACCF: Southern Antarctic Circumpolar Current Front, and sBdy: Southern Boundary front. Modern location of ODP sites (1090, Agulhas Ridge; 689, Maud Rise and 748, Kerguelen Plateau) used for reconstructions in this study. ODP = Ocean Drilling Program.**

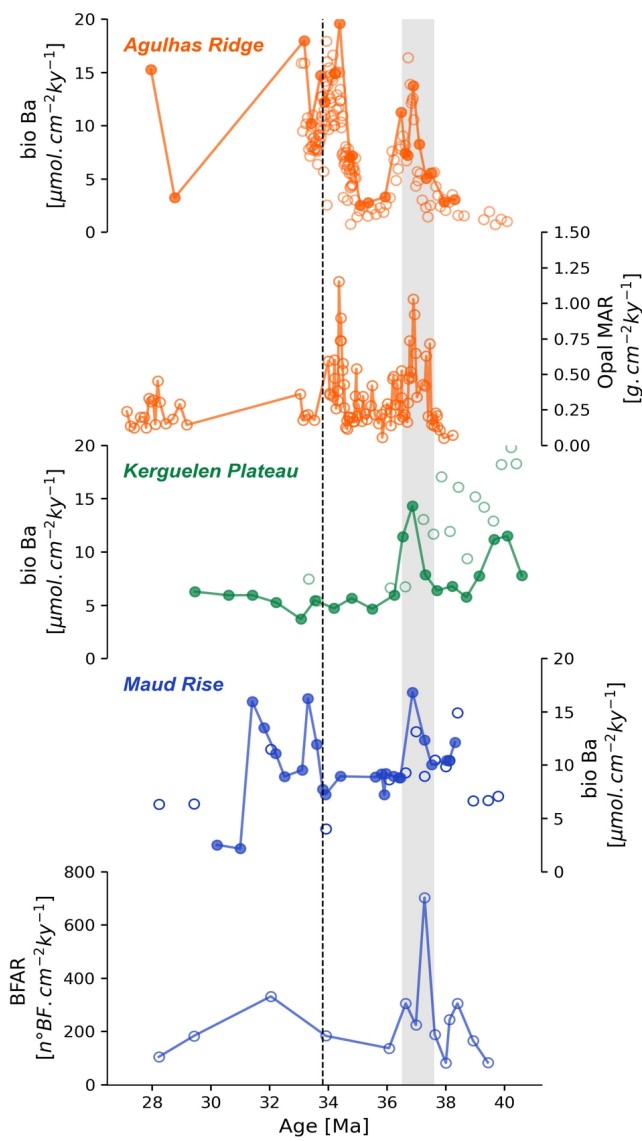

**Figure 2: Paleoprouctivity proxies vs Age (Ma) for Agulhas Ridge (ODP Site 1090, in orange), Maud Rise (ODP Site 689, in blue) and Kerguelen Plateau (ODP Sites 748, 744, in green). Solid circles are new biogenic barium accumulation rate (bio-Ba, µmol cm⁻² kyr⁻¹) data of this study, open circles from prior literature (Agulhas Ridge data from Anderson and Delaney, 2005; Maud Rise data from Diester-Haass and Faul 2019; Kerguelen Plateau data from Faul et al., 2010). Site 1090 opal MAR data are from Diekmann et al., 2004. Site 689 BFAR data are from Diester-Haass et al., 2019. Vertical bar identifies the E/O boundary (at ca 33.8 Ma). Shaded area encompasses the late-Eocene productivity event.**

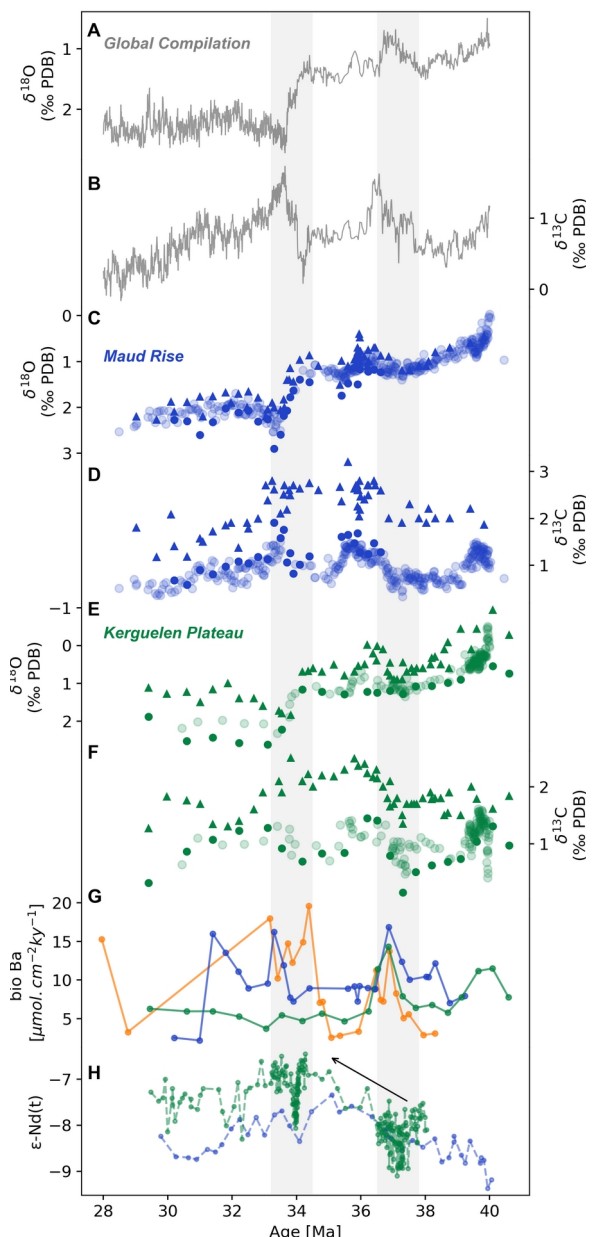

**Figure 3: Multiproxy records from the late Eocene and early Oligocene. Global compilation of oxygen and carbon stable isotopes (from Westerhold et al., 2020). New generated oxygen and carbon benthic foraminiferal isotopes data (solid circles) and fine fraction (<45μm) (solid triangles), and previously published oxygen and carbon stable isotopes (shaded circles, from Mackensen and Ehrmann, 1992; Diester-Haass and Zahn, 1996; Bohaty et al., 2003) from Atlantic Southern Ocean (Maud Rise) ODP Site 689 (in blue) and Indian Southern Ocean (Kerguelen Plateau) ODP Site 748 (in green). PDB is PeeDee Belemnite carbonate reference. Compilation of εNd data obtained from fossil fish teeth for the Atlantic Sector of SO (Maud Rise, in blue, Site 689), and for the Indian sector of SO (Kerguelen Plateau, in green, sites 738 and 748) (Scher and Martin, 2004, 2006; Scher et al., 2014; Wright et al. 2018). Shaded area identifies E/O boundary at ca 33.8 Ma and productivity event at ca 37 Ma. Note inverted y-axis scales for oxygen and Nd isotopes.**

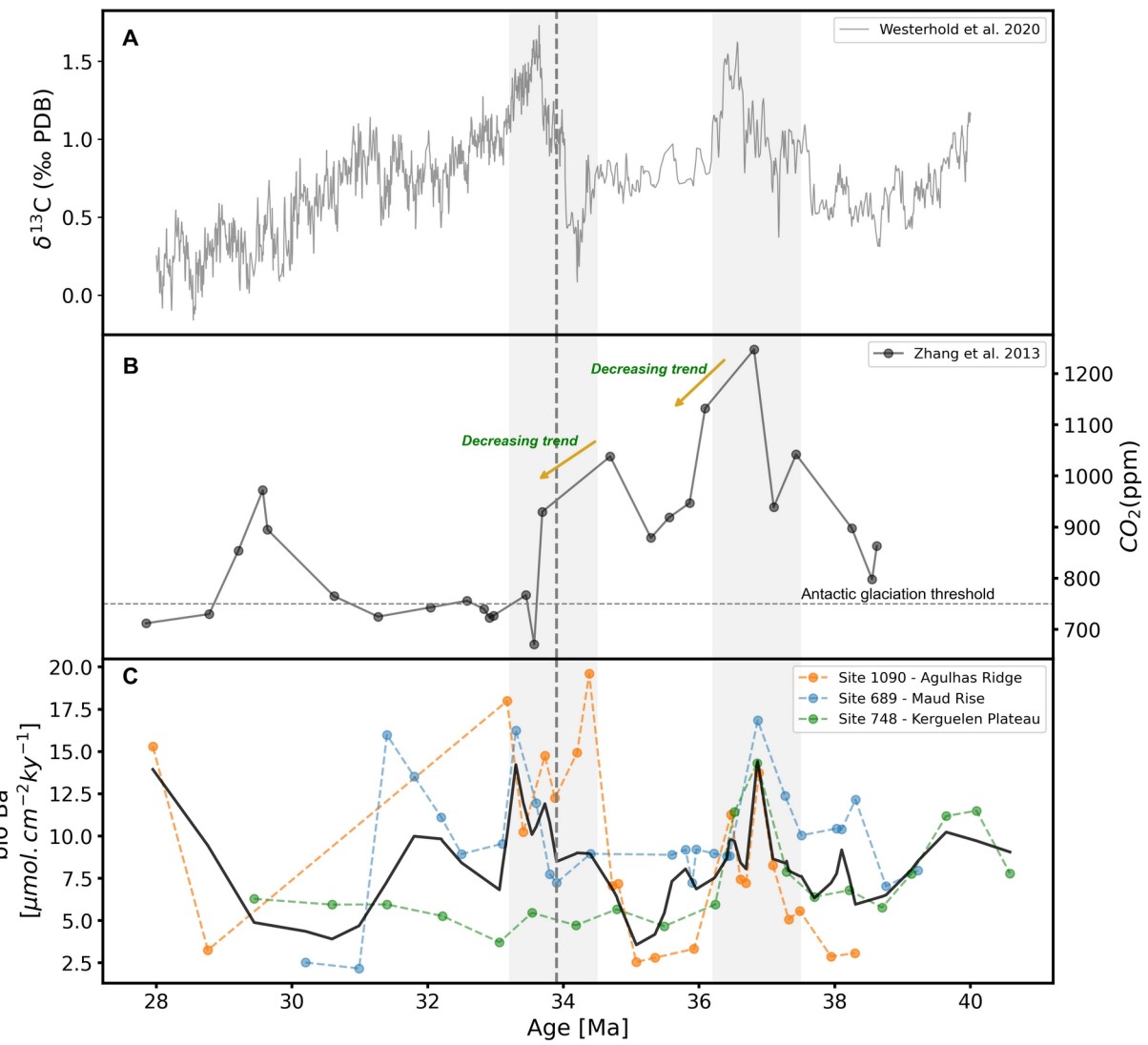

**Figure 4: Comparison between (a) a global compilation of carbon stable isotopes (from Westerhold et al., 2020), (b) alkenone-based atmospheric $p$CO$_2$ record (from Zhang et al., 2013) and (c) biogenic Barium (bio-Ba) export productivity proxy (this study). Antarctic glaciation thresholds (approx. 750 ppm) (from climate model, DeConto et al. 2008) is marked by a dashed line. Shaded areas encompass the late-Eocene and early-Oligocene high productivity intervals.**

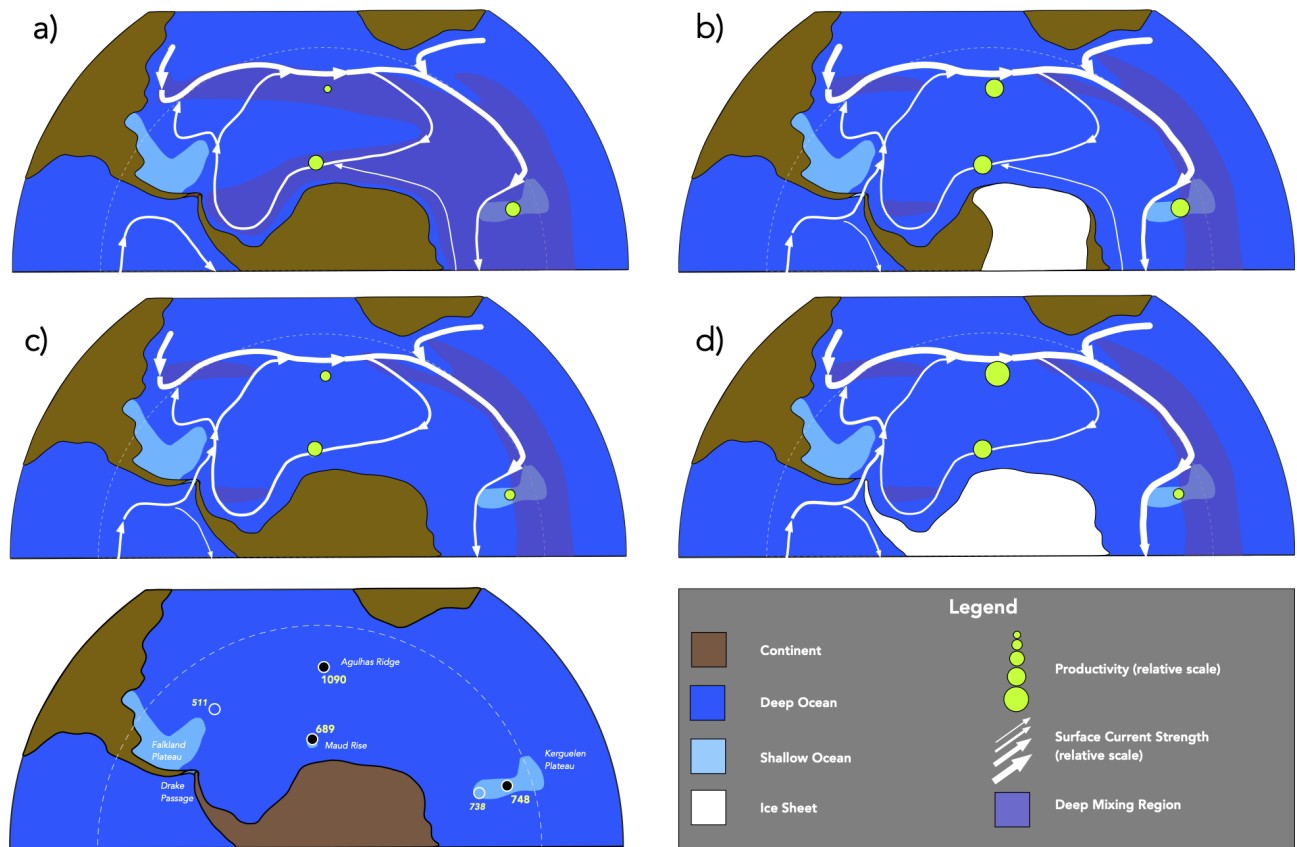

**Figure 5: Interpretive scenario of paleoceanographic change in the late Eocene to earliest Oligocene Southern Ocean. Base map, circulation patterns and extent of deep mixing regions largely after Toumoulin et al. (2020), ice sheet extent at 38 Ma after models in Van Breedam (2022). Productivity values based on results of this study, shown in relative scale. Note general trend towards higher productivity values, and within this, higher productivity, focussed near proto-ACC, during intervals with inferred ice sheets.**