# Peer review of "Late Eocene to early Oligocene productivity events in the proto-Southern Ocean and correlation to climate change"

_EGUsphere, 2023_

## Author Comment (AC2)

**Response to Referees Comments**

In blue, regular: reviewer comments
*In black, italic: authors reply*

**Referee #1: Prof. Peter Bijl**

Dear Prof. Peter Bijl,

Thank you very much for your consideration and very constructive feedback on our paper. We have made several changes based on your comments and recommendations. We have detailed our responses below.

All the best,

Gabrielle Rodrigues de Faria, David Lazarus, Johan Renaudie, Jessica Stammeier, Volkan Özen, Ulrich Struck.

**Introductory remarks:**

The authors have generated biogenic barium and stable carbon and oxygen isotope records at three drill sites in the Southern Ocean spanning the late Eocene to early Oligocene. By intricately comparing the high-resolution datasets, with age models that were constructed in previous work, the authors recognized 2 episodes of increased export productivity, in the lower-Priabonian and around the Eocene-Oligocene transition. The authors correlate these to episodes of cooling and decline in atmospheric CO2, and conclude that the records express phases of enhanced carbon drawdown by the Southern Ocean, storage of that carbon and through that explains the CO2 decline. The stratigraphic correlation of the sites seems robust, although since the work on the age models was done elsewhere, there is very little that the authors provide in this paper to verify that. It would help the reader to have a (supplementary) figure showing those age models. I am not an expert on the methods used by the data seems of high quality, generating using standard methodologies and are of high enough resolution to detect the trends. The paper is overall well written (occasionally the wrong use of the word "if" confused me) and structured. My main concern is on the interpretation step from export productivity (which in my view has been clearly demonstrated) to CO2 storage and through that 1 million-year CO2 decline.
In conclusion, I value the records, and find the consistency between the records definitely fascinating, but I am not convinced of the evidence put forward by the authors that link export productivity in the Southern Ocean upwelling zone to atmospheric CO2 decline without carbon cycle box modelling exercises that marries all observations. I will substantiate this further below.

*Response: Our response to age model comments is given here; other comments are below. Dr. Bijl comments that the age models were apparently developed 'elsewhere' and are not adequately documented. We will add a more explicit statement that the age models were indeed developed by us, albeit using only published stratigraphic data. This was done in the context of our much more extensive age model work (ca 500 deep sea drilling holes) for our NSB/Neptune database. We do already provide the age models as plots in the SOM, and also a link to the NSB/Neptune database, where one can normally*

*download the full, synthesized data used to develop them. We regret however to report that the link has not been functional for a few weeks as the Museum für Naturkunde entire IT infrastructure has been locked up due to a ransomware attack. Although the primary data in NSB is publicly backed up at Zenodo ([https://zenodo.org/records/10063218](https://zenodo.org/records/10063218)), creating the synthesized data from the underlying database tables requires some effort. We will thus add tables of the synthesized stratigraphic data for each age model created in our study to the SOM, although we hope our IT systems, and NSB, will be up and running by the time the ms is formally published.*

**Comment: From export productivity to atmospheric CO2 decline**

I have some serious concerns about the way the authors interpret their local export productivity records to ocean carbon drawdown and atmospheric CO2 decline. Indeed, during the Pleistocene, the ocean plays a large role in the glacial-interglacial variability of atmospheric CO2 concentrations through ocean carbon storage, but on longer time scales, atmospheric and ocean carbon reservoirs are in equilibrium (which explains the long-term Pleistocene CO2 stability). Regions of upwelling bring the excess deep-ocean carbon back into the atmosphere.

In fact, the argumentation by the authors demonstrates the complexity: they invoke an increase of upwelling to explain the stimulation of export productivity, but omit to confess that upwelling areas are vast ventilators of deep-ocean carbon into the atmosphere. Invoking increased upwelling to explain the extra carbon export complicates the interpretation of the pCO2 decline. In the modern Southern Ocean, it is not the upwelling areas with high primary productivity that constitute the carbon sink: it is the subantarctic zone, where surface waters low in DIC flow northwards and adsorb atmospheric CO2, after which they sink in intermediate water formation. The zone of maximum export productivity is in an upwelling zone and therefore a zone of CO2 flux out of the ocean into the atmosphere (see, e.g., the work by (Egleston et al., 2010; Sabine et al., 2004)).

Coming back to the timescale issue, (Sluijs et al., 2013) has demonstrated the complexity there. It showed that on timescales of 0.5-1 million years it is really difficult to induce carbon cycle perturbations, but at least it is doubtful whether just export productivity to the deep sea can do the trick. According to Sluijs et al., 2013, carbon deposition on the continental shelves plays an important role, but in the late Eocene and across the EOT, ice sheet formation creates progressively less flooded continental shelve area. In any case, it seems like OC burial records are needed to seriously affect atmospheric CO2 decline on these time scales. The thought experiment (around line 400) that combines assumptions from the modern Southern Ocean that we know are not realistic for the Eocene (deep ocean ventilation, ocean flow strength) is grossly inadequate and doesn't do justice to the knowledge we have on late Eocene ocean conditions. It is actually quite crucial that the authors demonstrate excess carbon burial to relate their export productivity trends to carbon removal from the exogenic carbon pool.

The authors suggest that knowledge on the carbon cycle is limited to get full understanding (lines 385/386. But this bypasses a slur of knowledge obtained from carbon cycle box modelling such as LOSCAR (Zeebe, 2011) as was also used in Sluijs et al., 2013. This box model takes into account the full carbon cycle, upwelling as well as ocean carbon uptake, alkalinity and CCD trends in all ocean basins, and includes stable isotope tracers to simulate the effects on deep-sea carbon isotopes. I would claim to say that without such a modelling exercise, the translation from one element of the carbon cycle (in this case export productivity) to atmospheric CO2 is impossible to make.

**Response:** *To summarize, both reviewers acknowledge that most of the main points of the ms are largely acceptable, albeit with numerous suggestions for improvement or clarification, which we individually address below. The reviewers accept the two main points of our ms - documenting the existence of intervals of increased export productivity in multiple locations in the proto-Southern Ocean (SO) during the late Eocene; and that these productivity peaks are largely coincident with published estimates of pCO2 change, shifts in carbon cycle proxies such as ∂13C, and glacial intervals on the Antarctic continent.*

*Most of the critical commentary of both reviewers focused on the ms calculations and discussion of quantitative links between our export productivity results and the drawdown of atmospheric pCO2. As this is the main focus of critical comments and is repeated in several places in the individual comments of both reviewers, we give our general response here, and refer most of the individual comments of the reviewers on this theme back to this in subsequent replies below.*

*Both reviewers express concerns that we have not adequately analyzed or discussed this very complex topic, and have not proved in our ms that the productivity pattern was in fact the actual driver of pCO2 change. We fully agree with both reviewers, as this was not the intent of this part of the ms. The misunderstanding clearly rests on poorly worded ms text and our failure to properly explain the goal of this ms section at the beginning. Our intention in this section was to make a quick 'sanity check' type of calculation. Specifically, aiming to determine what the total carbon sequestration maximally might have been, without considering any compensating factors that would have reduced the amount sequestered. We are fully aware that this is highly unlikely. But, if the maximum magnitude calculated was dramatically lower than the actual observed pCO2 change, then we could immediately disprove the idea that our productivity results could potentially be related to the pCO2 drop in the late Eocene. In our opinion, the calculations, despite being crude, show that the hypothesis passes this initial test. We completely agree that to actually prove a direct link between the productivity results and pCO2 drawdown requires extensive additional studies, including modelling studies, much better parameterisations of ocean characteristics (e.g. among many, warm water column degradation of Corg), additional data on e.g. organic carbon sequestration in sediments etc., all of which are well beyond the scope of our ms.*

*In light of these considerations, we are therefore open to rewrite this part of the text, making clear that we are not attempting, or able, in our ms, to prove any link between productivity increases and pCO2 drawdown. Our manuscript can show that there are productivity increases, that they temporally correlate to published records of pCO2, carbon cycle and climate change, and that the magnitude of the productivity change was substantial. The actual impact it had is however fully open and will require future research. We would in fact like to incorporate, in paraphrased form, some of Dr. Bijl' comments on the type of work needed here, assuming he has no objection.*

*As an alternative, we could consider deleting all calculations and discussion of the potential of productivity to alter pCO2. We do not however prefer this option, as our ultimate goal of studying productivity changes is to contribute to understand the causes of pCO2 change in the late Eocene, and via this, to improve our understanding of the Cenozoic climate change record. Our section is only a modest bridge from our results to these future studies, but we think an appropriate one to make.*

**Comment: The importance of bathymetry**

The location of the sites is not trivial, and I feel the authors must pay a little bit more attention to the local bathymetry at these sites. Bathymetry, however high, plays a huge role in steering local ocean flow, even up to the surface. For the Late Eocene, this was demonstrated in the high-resolution ocean model simulation by (Nooteboom et al., 2022). As such, the site record first and foremost local changes in oceanography, and depending on the bathymetric anomalies around, they are representative of what happens on a larger area. Many modern bathymetric highs have upwelling associated to them, because the high pushes deep ocean water upwards. Many ocean regions with bathymetric obstructions also are sensitive to small climate or oceanographic changes, as ocean fronts are unable to flow over them, so have to go around. Therefore, in most cases, close to bathymetric obstructions, oceanographic changes are strongly amplified compared to the forcing or the latitudinal average oceanographic changes. I just note that all three sites come from bathymetric highs, but are in this study used in a large extrapolation exercise to calculate whole-Southern Ocean carbon storage. I think that, given what we know of the amplification effect of local ocean change by bathymetry, this important point must be addressed in the paper, and particularly at an extrapolation exercise.

*Response: Thank you for your suggestion. We acknowledge that all three sites are located in bathymetric highs and the importance of bathymetry in oceanographic conditions. In our revised manuscript we will provide more detailed sites' descriptions and address in our discussion the potential impact of bathymetric anomalies on our findings. However, all sites are deeper than 1 km paleodepth, and thus unlikely to have created major, highly localized 'island' effects on surface water upwelling and productivity; particularly if, as the reviewers themselves suggest, the protoACC penetrated only to significantly shallower depths in the Paleogene. Thus the sites are in our opinion reasonably good representatives of oceanographic conditions for the regions of the ocean studied. Further, bathymetry remained the same over the study period, so the observed temporal variations in export productivity are unlikely to be related to changing bathymetry.*

**Smaller points of attention**

**Comment:**

Abstract: I categorically disagree with the way two consecutive theories for the trigger for the EOT "climate shifts" (I would write ice sheet formation, but ok) are presented as equal. They are not, and have never been. The gateway opening theory as primary trigger for AA glaciation from the '70s has been thoroughly refuted in the '90s, '00s and '10s by meticulous dissection of its argumentation with evidence from modern physics (Sloan and Rea, 1995), numerical modelling (Huber et al., 2004; Huber and Nof, 2006) and a lot of microfossil data e.g. (Bijl et al., 2011; Houben et al., 2019). At the same time that this hypothesis was refuted, the evidence for the role of $CO_2$ decline in explaining AA glaciation appeared (Deconto and Pollard, 2003a, b; Deconto et al., 2008). So these two hypotheses were never really competing, they were merely consecutive. Later studies have asked the question: "if gateway opening wasn't the primary trigger of AA glaciation, then what was their secondary role?" (Sauermilch et al., 2021), in line with the thinking since the '00s that $CO_2$ was the primary force. But that was new to compete with $CO_2$ having been the primary trigger. Compelling new evidence in support for the gateways providing the primary trigger for AA glaciation has not been presented since then. So, the driving mechanisms are not controversial, but surely we do need a better understanding

about how different environmental parameters relate to each other (line 15). The more important question surrounding environmental changes around EOT revolve around forcings versus response, particularly in the Southern Ocean: given that $CO_2$ forced climate (and also ocean) cooling, and Antarctic glaciation, which in turn could have triggered further ocean changes, which part of the oceanographic changes observed in records is forced by the $CO_2$ decline and which part is the consequence of the continental-scale ice sheet buildup, perhaps though atmospheric feedbacks (Houben et al., 2019).

*Response: Dr. Bijl is fully correct in that the original 'tectonics OR CO2' view has been replaced, and it is clear that CO2 played a key role. However not all workers are convinced that tectonics played at most a minor role - as cited in the ms introduction.  We also will add more explicit citation and review of the important concept of how tectonics may have driven the CO2 change, via the effect on Southern Ocean circulation and ocean productivity (e.g. Egan et al. 2013, EPSL).*

**Comment:** Lines 27/28: These papers do not really demonstrate that. Perhaps the study that really tries to quantify the importance of Cenozoic climate boundaries is (Westerhold et al., 2020)

*Response: We agree and added Westerhold et al., 2020 as a better reference for this sentence.*

**Comment:** Lines 55/56: What is mostly unclear is that this threshold is model-dependent and strongly dependent on boundary conditions (Gasson et al., 2014)

*Response: This is a valuable observation, the point has been added in the paragraph.*

**Comment:** Lines 61/62: A lot of more recent literature suggests it is not - well, it depends on how you define an ACC. It triggered possibly some kind of circumpolar flow, but the onset of modern-like strength was strongly stalled uptil about 10 million years ago (e.g., (Evangelinos et al., 2020; Evangelinos et al., 2022)

*Response: Text has been reworded to make the sentence more clear.*

**Comment:** Line 75: Not the development of all fronts: There always was a subtropical front as the boundary between the subtropical and subpolar gyres (see all modelling studies). The authors picture it in their Figure 5 quite clearly.

*Response: The text has been reworded to make the sentence more clear.*

**Comment:** Lines 126/127: the direction of flow of the ACC and in part strength is driven by westerly winds, but its exact flowpath is governed by bathymetry (not unimportant given that the sites you present data from are onto those main obstructions)

*Response:* *Text has been reworded to emphasize the importance of bathymetry in the flow path of the (modern) ACC. But see the prior comment on the limitations of bathymetry on our analyses above.*

**Comment:** Lines 153/154: very gradual. For most of the time, Australia still blocked the ideal flow path of the ACC (Hill et al., 2013)

*Response:* *In the preceding sentence, we have emphasized that removing the barriers is essential for the gradual development of the circumpolar flow. This includes Australia blocking the ideal flow of the ACC, as pointed out by Hill et al., 2013.*

**Comment:** Lines 161/162: Although this is already a long list, I feel that some crucial papers are missing here, mostly because they represent a leap forward in the approach. All model simulations here attempt to simulate equilibrium ocean circulation in fully coupled climate models, but these simulations come at the expense of spatial resolution. That this seriously impacts results of ocean flow has been demonstrated by Nooteboom et al., 2022 who simulates late Eocene ocean flow in a high-resolution simulation, that resolved Eddie flow. Secondly, as the authors rightfully say before, tectonic changes are really important, and Sauermilch et al., 2021 demonstrated the oceanographic effect in several steps across the EOT.

*Response:* *We appreciate your input and agree. The references mentioned will be included in the revised manuscript.*

**Comment:** Lines 290/291: Very old data, some of which are now deemed unreliable (Pearson and Palmer). Please refer to the paleoCO2 compilation for the most recent work.

*Response:* *All pCO2 data used in our study are listed in the Supplemental material, and the data from paleoCO2 compilation have been included in our study.*

**Comment:** Lines 375/376: An important study that precedes the current study is that of Houben et al., 2019, that documented carefully the surface oceanographic conditions preceding the onset of AA glaciation in the Southern Ocean. This paper concluded a spinup of the ocean flow as evidenced by widespread glauconite formation in the Southern Ocean, as well as microplankton evidence.

*Response:* *The reference will be added to our ms.*

**Referee #2: Anonymous**

Dear Anonymous reviewer,

We appreciate your insightful comments and detailed feedback. We are committed to addressing the points raised in our revised manuscript. We have detailed our responses below.

All the best,

Gabrielle Rodrigues de Faria, David Lazarus, Johan Renaudie, Jessica Stammeier, Volkan Özen, Ulrich Struck.

**Review of Late Eocene to early Oligocene productivity events in the proto-Southern Ocean as drivers of global cooling and Antarctica glaciation by Rodrigues de Faria et al.**

In this work the authors present an extensive overview of Southern Ocean oceanographic changes for the late Eocene and Eocene-Oligocene Transition. The focus of their work is disentangling the effects of potential increases in primary productivity associated with changing current systems and associated CO2 drawdown from other (e.g. tectonic) effects that have been hypothesised to explain or form prerequisites for the establishment of major ice caps on Antarctica.

They aim to achieve this by combining new and published productivity data (Ba, carbonate stable isotopes, biosilica MAR) and published data on pCO2, eNd. The paleo-productivity proxies show a robust signal, which signals an increase in Southern Ocean productivity in two intervals preceding and overlapping with the EOT. Finally, the authors attempt to link these intervals of increased productivity in the SO to global trends in 13C and pCO2.

Overall, I found the hypotheses tested the manuscript interesting and overall it is reasonably well-structured and well-written. The data-aspect (new Ba-bio data, carbonate isotopes) and the trends these data show are interesting and seem robust. The correspondence of the two intervals of increased productivity to global d13C increases is intriguing and the overall picture on EOT oceanography and productivity patterns obtained from the new and revisited data as well as the updated age-models is well suited to the journal. I am however concerned about validity of the assumptions that support the search for carbon sinks and the subsequent steps taken to test whether C drawdown through increased export productivity or ocean current restructuring was an important factor. This stems from two main points of uncertainty – first, whether the trends in pCO2 required overall CO2 drawdown and second whether a regional productivity signal actually reflects CO2 drawdown on a global scale.

**Carbon burial – changing balance vs increase/decrease**

Although perhaps not directly related to the data presented here, a few choices made in the manuscript regarding interpretation of the previously generated pCO2 and 13C records warrant close attention. The data is somewhat ambiguous as to whether the periods of increased productivity are associated with enhanced carbon drawdown. This is particularly relevant because the pCO2 records from alkenones and 11B show different signals and 13C can be interpreted in two different ways. First is the scenario preferred in this study; the alkenone-based pCO2 from Zhang et al., (2013) shows a decrease while benthic 13C

increases – this could indeed be consistent with an overall increase of organic (13C-depleted) carbon burial. In a second scenario, taking 11B as the preferred pCO2 proxy, very limited change occurs coeval with the 13C increase, which would suggest an increase in the relative importance of 13C-depleted carbon burial but no overall increase. Much progress has been made in understanding the origin of the alkenone fractionation signal (cell size, growth effects etc, see e.g. Stoll et al., (2019) and Zhang et al., (2020) and references therein) since the publication of the Zhang et al. (2013) record and I feel using only this record at face value might be difficult to justify. This is particularly pertinent as the 11B record may be taken to suggest that there is no pCO2 decrease to explain at all.

*Response: We have shown all pCO2 data in the supplemental material, Figure S7 and Table S2. Our decision to show in the ms the dataset from Zhang et al., 2013 is justified in the ms, and we stand by it here, in the fact that the data come from a single site with a consistent methodology that spans the studied time period. This choice ensures a more uniform and reliable representation to compare with the trends in our data. Combining temporally patchy data with different, and poorly known systematic biases does probably give a less biased view of the absolute magnitude and long-term trend, but at the cost of additional noise and risk of short-term artefacts that obscure important shorter term events.*

**Mechanisms of C drawdown associated with ocean circulation**

Two general pathways are traditionally (and here) considered; uptake of C in certain ocean regions through sea-air disequilibrium (solubility) and uptake of C in tissue (productivity) followed by burial in sediments. In general terms both can affect atmospheric C but they generally operate on different time-scales and have very different overall potential to store C permanently. In addition, these processes may have functioned different from present-day due to changes in boundary conditions so that the extrapolation based on modern-ocean characteristics may not be warranted. In my opinion these distinctions and complexities are not sufficiently addressed in the manuscript and I encourage the authors to deepen their discussion on these subjects.

For time-scales that exceed kyrs, disequilibrium may induce a change in CO2 only through a change of the ocean's potential to store C, so that more/less C is stored in the atmosphere relative to the ocean. This implies a permanent oceanographic reorganisation could indeed produce a (small) drop in atmospheric pCO2. The sign of the solubility effect is however difficult to predict and, given that the global d18O signal suggests a period of warmer deep water conditions for the 37 Ma interval, I could argue "solubility" or the potential for the whole ocean to store more C actually decreases. What makes projecting the effects of Southern ocean solubility around and before the EOT even more complicated is that there is likely a shift in the balance between upwelling and downwelling waters. Upwelling waters are often DIC-rich, so that they can release C to the air, partially offsetting the drawdown effects that may happen further poleward (e.g. Lauderdale et al., 2017).

For the productivity effect, which is the focus of this work, a few more steps are required before it can be judged whether the effect is as important as it may appear. First and most important, I think the authors can increase confidence in their claims by including a (stepwise) method of how export productivity is converted to C burial, as export productivity itself is not the ultimate sink. Second, various physico-chemical parameters of the Southern Ocean at the time need to be considered in this calculation – there is a range of studies that have shown, for example, that the positions of the fronts and temperature

regimes for this period were very different from the present-day. The positions of the fronts largely determines whether the productivity estimates were representative of the Southern Ocean at the time and how aspects of the modern ocean, such as export productivity to orgC burial estimates, can be incorporated in the extrapolated EOT dataset. The temperature regime impacts orgC burial efficiency through remineralisation (e.g. John et al., 2013); remineralisation is more rapid and efficient in warmer waters. There is abundant proxy evidence that the late Eocene/early Oligocene Southern Ocean was much warmer than present-day (e.g. Kennedy-Asser et al. (2020) and references therein). The evidence for such effects implies the structure of the biological pump in the Southern Ocean was likely quite different from present-day and needs to be considered when relying on water-column proxies for productivity to infer (trends in) C burial.

*Response: See the general comment on this theme near the top of the response to Reviewer 1.*

**Specific comments (by line number)**

**Line 43** 'long timescales' is a rather vague statement and depending on which effect is considered may not be valid in geological context. OrgC burial and solubility are reasonably fast feedbacks, whereas silicate weathering might not be.

*Response: See comment about tectonic controls on ocean productivity above.*

**Line 53-57** I think atmospheric pCO2 estimates for this period have been adjusted to lower values over the past few years - (e.g. Anagnostou et al., 2016; Zhang et al., 2020), may need updating.

*Response: Thank you for bringing this to our attention. We have updated the manuscript according to the new estimates by Anagnostou et al., 2016.*

**Line 76-79** This leaves a few things unsaid that I think are important to mention (see also major comments above). Particularly the conversion of export productivity to C-burial is not straightforward.

*Response: See the general comment on this theme near the top of the response to Reviewer 1.*

**Line 86-88** OK – but here you intend to use these same sites; can you support the assumption that they are (at least collectively) representative of the conditions in the Southern Ocean?

*Response: Many of the studies mentioned focused on individual sites, each with variations regarding proxies used, methodology and age models. Our approach differs by using all 3 sites, applying consistent proxies and methodology across all sites and updating the age models. By utilizing multiple sites, in different sectors of the Southern Ocean; both near, and well to the south of the (paleo) ACC, and ensuring methodological consistency, our study provides a broader and more representative condition of the Southern Ocean than the cited previous studies. We agree that it is not truly comprehensive and have already so stated in the ms ('Limitations…' section).*

**Line 94** See also previous comment on export productivity – Ba-bio records the signal relatively high in the water column and may be considered a less direct proxy for orgC burial than for example benthic foram accumulation (or actual orgC accumulation).

*Response:* BFAR is indeed important, we are working on this to address this issue by combining new data with BFAR from previous studies. This will be included in a new ms when the work is finished. Corg in sediments is also important, but beyond the scope of our current research. We do already note that more such data is needed in the 'Limitations…' section.

**Line 180-183** Somewhat odd end to the introductory paragraph – an ending with the main aims and approach of the study also included (as done at the end of §1.1) might be a better fit. Perhaps worth considering a small change in structure here.

*Response:* Thanks for spotting this! A bridge sentence will be added.

**Line 208** Is there a specific reason not to use GTS2020? (may not differ much for this period)

*Response:* It does not differ much from GTS2012. Much of the published data has been interpreted with the 20212 scale, and all age data in the NSB system is also referenced to this scale. It is thus more convenient to use GTS2012.

**Line 210** Delete the "210"; Leg number is a little confusing here

*Response:* Agree. The leg number has been removed in our revised manuscript.

**Line 238-241** Sentence reads as if to suggest that changes on scale of 5 Myr cannot be detected; is that correct? It seems to be an unexpectedly large error.

*Response:* We appreciate your keen observation. It is a typo error with the '.' present but attached to the wrong symbol. The correct scale is 0.5 Myr.

**Line 258** Check if the mention of (only) HCl here is correct; HCl does not dissolve silicates.

*Response:* Thank you for your detailed review of our paper. We have corrected the error in our revised manuscript. The samples' preparation included fusions with N2O2 and HCl, following the method of Bokhari and Meisel, 2017.

Bokhari S.N.H. and Meisel T.C. Method development and optimisation of sodium peroxide sintering for geological samples. Geostandards and Geoanalytical Research, 41, 181–195, 2017.

**Line 273** The first mention of Nd isotopes; might be good to briefly mention / introduce this proxy before this point

*Response:* Agree. We have introduced the Nd isotope proxy earlier in the Introduction to provide context before mentioning the data compilation.

**Line 301-305** See also comments above on alkenone pCO2 estimates

*Response: This has already answered above.*

**Line 389-392** The extrapolation from modern ocean parameters and oceanographic structure to a past state needs to be discussed in more detail to be able to judge whether such an extrapolation is warranted

*Response: See general comment on this theme near the top of response to Reviewer 1.*

**Line 398-399** As transported orgC is not the same as buried orgC; is this somehow converted or is it simply assumed to scale linearly?

*Response: See previous response.*

**Line 398-406 & table 2** This all seems dependent on the orgC flux not to be compensated by reduced orgC burial elsewhere – I appreciate that this is one aspect that will always remains uncertain but I think important to note regardless. In a similar vein, out of the total orgC burial flux, the Southern Ocean is likely a small component – most open ocean settings are not exceptionally important as a C sink, compared to continental shelf and deltaic sediments. More important for this work, it also remains unclear to me how C burial is converted to atmospheric $CO_2$ loss; what steps are included here? Are you using the total amount buried (max 1750 PgC) and removing that from the atmosphere or atmosphere-ocean reservoirs? Is there repartitioning between reservoirs?

*Response: See the general comment on this theme near the top of the response to Reviewer 1.*

**Line 419-430** The framing/reasoning in the paragraph on 13C is somewhat confusing – for example, when suggesting a reasonable fit with global records I was not expecting to see such an offset (13C peaks at ca. 36 Ma rather than 37 Ma) between the global compilation and your Atlantic and Indian ocean benthics. This also implies the highest productivity intervals do not seem to be aligned with the highest 13C and perhaps not even with increasing 13C – but how these are related exactly is difficult to tell as the records are in two different plots. If indeed these are not aligned – how certain are you that the trends in 13C represent or reflect increased (regional) productivity and how does the increased productivity produce a difference in 13C trend in bottom waters or is this only the result of changing circulation? More guidance is needed to follow the reasoning here.

*Response: We agree there is a certain degree of offset. This could well be due to the limited number of datapoints in each time series or residual age model errors; or true offsets due to the regionally varying co-action of other (unknown) mechanisms. The correlations, if not perfect, are in our opinion rather good given the limitations of the data. Hopefully, future studies with more data and even better chronology will be able to better resolve this issue.*

**Line 537-539** I think solubility effects (if included) should be introduced at an earlier stage (for example in context of Pleistocene glacials)

*Response: We will add an earlier comment as suggested.*

**Line 539-541** Here the 'could have' is key – I do not know if there is any proxy evidence to support or refute this hypothesis

***Response:*** *We acknowledge the uncertainty surrounding proxy evidence and modelling studies to definitely support or refuse this hypothesis. Thus we flagged the speculative nature by the use of 'could have'.*

**Line 553-555** The global effect may be negligible (any CO2 taken up near Antarctic is simply outgassed elsewhere) unless the capacity of the whole ocean to store DIC is increased.

***Response:*** *See the general comment on this theme near the top of the response to Reviewer 1.*

**Line 555-558** The statement on upwelling and CO2 drawdown in cold waters is likely to be incorrect because the upwelling waters are almost always oversaturated wrt CO2, even in cold regions. Most upwelling zones are net degassing and normally considered CO2 sources.

***Response:*** *Thank you for pointing out the need for clarification. Nutrients supplied by upwelling are a key component of the biological carbon pump, and thus our statement is broadly correct. We say that upwelling increases productivity, which, in turn, has the potential to contribute to the drawdown of CO2. We will ensure to revise the text to eliminate any possibility of misinterpretation. Additionally, in our revised manuscript, we will also discuss upwelling areas as ventilators of deep-ocean carbon into the atmosphere.*

**Figure 2-3**, the combination of these data is important for understanding the reasoning in the manuscript and I would welcome a version of Fig. 3 where at least one (preferably all) of the new productivity reconstruction data are included to illustrate similarities and differences in trends.

***Response:*** *Thank you for your valuable suggestion. In our revised manuscript, we will update Figure 3 incorporating our new productivity data to better illustrate similarities and differences in trends.*